# CAUCHY-SCHWARZ FAIRNESS REGULARIZER

## ABSTRACT

In this paper, we propose a novel approach to fair machine learning, the *Cauchy-Schwarz fairness regularizer*, which minimizes the Cauchy-Schwarz divergence between the prediction distribution and sensitive attributes. While existing methods effectively reduce bias as indicated by low values on specific fairness metrics, they frequently struggle to achieve a balanced performance across various fairness definitions. For example, many approaches may successfully attain low demographic parity yet still demonstrate significant disparities in equal opportunity. Theoretical studies have shown that the Cauchy-Schwarz divergence provides a tighter bound compared to the Kullback-Leibler divergence and gap parity, suggesting its potential to improve fairness in machine learning models. Our empirical evaluation, conducted on four tabular datasets and one image dataset, demonstrates that the *Cauchy-Schwarz fairness regularizer* achieves a more balanced performance across fairness metrics while maintaining satisfactory utility. It outperforms existing fairness approaches, providing a superior trade-off between fairness and utility. In addition, the *Cauchy-Schwarz fairness regularizer* is a versatile, plug-and-play fairness regularizer that can be easily integrated into various machine learning models to promote fairness.

## 1 INTRODUCTION

Machine learning models are increasingly adopted in high-stakes decision-making scenarios, such as credit scoring (Petrasic et al., 2017), the job market (Hu & Chen, 2018), healthcare (Grote & Keeling, 2022), and education (Bøyum, 2014; Kizilcec & Lee, 2022). Despite their success, these models are often prone to generating prediction disparities among different demographic groups, including genders, ages, skin colors, and regions, particularly when no interventions are introduced during the training process (Mehrabi et al., 2021; Dwork et al., 2012; Barocas et al., 2017). Such biased algorithms can have detrimental impacts on individuals' lives, especially for disadvantaged groups. This inherent bias in the data complicates the pursuit of fairness in machine learning models (Jiang & Nachum, 2020). Consequently, the growing concern over fairness has garnered significant attention from researchers, who are striving to achieve equitable predictions across demographic groups based on various fairness notions (Hsu et al., 2022; Chai et al., 2022; Reddy et al., 2021).

Many debiasing methods incorporate a fairness regularizer that aims to minimize the differences in prediction distributions across various sensitive groups. These prediction distributions are typically assessed using metrics such as gap parity, Kullback-Leibler (KL) divergence, and the Hilbert-Schmidt Independence Criterion (HSIC). While these methods can effectively enhance performance on certain fairness metrics, they often fail to maintain a balanced level of fairness. For instance, as shown in Figure 1, the DP regularizer successfully achieves low demographic parity (DP), indicated by the closely aligned prediction distributions for female and male groups. However, it does not significantly improve equal opportunity (EO), as evidenced by the considerable disparity between these two distributions. This finding is further corroborated by the T-SNE plots, which illustrate that the embeddings for the female (blue points) and male (red points) groups are indistinguishable across all data points, particularly for $Y = 0/1$. In contrast, for the positive class data points ($Y = 1$), the embeddings exhibit a more discernible pattern, with the blue points clustered in a specific area (circled) rather than being evenly distributed, as depicted in the T-SNE plots for all classes.

This indicates that training a model solely to optimize its objectives is insufficient for achieving fairness in test data. Previous studies have shown that a machine learning model that achieves a 0 disparity in training may still fail to maintain low disparity on test data, indicating an inability to

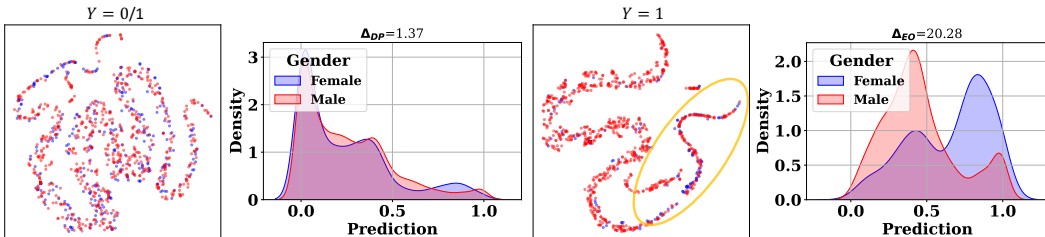

Figure 1: From left to right: (1) Prediction distribution of *all classes*; (2) T-SNE visualization of embeddings for samples from *all classes*; (3) Prediction distribution of *class 1*; (4) T-SNE visualization of embeddings for samples from `Adult`, and the sensitive attribute is gender. The blue points represent samples with sensitive attribute 0, while the red points represent samples with sensitive attribute 1.

generalize fair awareness. The fundamental issue is that it is crucial to find an appropriate method for measuring distribution divergence. For instance, the DP regularizer is implemented by calculating the difference between the *average* predictions or probabilities. However, relying solely on the distance between mean estimators does not accurately capture distribution divergence; two distributions can differ significantly even if their mean estimators are identical. A similar phenomenon was observed with the fairness regularizers of KL divergence and HSIC in our preliminary analysis Section 4.3. This observation highlights the need for more advanced divergence measurement techniques.

Previous studies have demonstrated that the Cauchy-Schwarz divergence offers a tighter theoretical bound than the KL divergence and performs well in domain adaptation for image datasets. Our theoretical analysis demonstrates that the Cauchy-Schwarz divergence is particularly effective for improving fairness in comparison to existing fairness regularizers. In light of this, we propose a new fairness regularizer based on the Cauchy-Schwarz divergence for fair machine learning. We summarize our contributions as follows:

- We introduce the Cauchy-Schwarz divergence to fair machine learning and present a novel regularization-based method.
- We elucidate the relationships between the Cauchy-Schwarz regularizer and other fairness regularizers, emphasizing its superior effectiveness in debiasing.
- Our experimental results, obtained from four tabular datasets and one image dataset, validate the efficacy of the proposed Cauchy-Schwarz regularizer in achieving fairness across multiple fairness notions simultaneously.

## 2 PRELIMINARY

In this section, we establish the foundational concepts for our study. We start by exploring the notion of fairness in machine learning, including the relevant notations. Next, we provide an overview of general fairness-aware machine learning methods. Finally, we introduce the Cauchy-Schwarz divergence and discuss its benefits in reducing bias.

### 2.1 FAIR MACHINE LEARNING

There are various notions of fair machine learning, with group fairness being one of the most extensively studied concepts in recent years. In this study, we focus specifically on group fairness and introduce additional notions in Appendix F.2.

Group fairness seeks to ensure that machine learning models treat different demographic groups equitably, where groups are defined based on sensitive attributes such as gender, race, and age. This concept is rooted in the notion of statistical parity, as discussed in prior studies (Feldman et al., 2015; Zemel et al., 2013). Specifically, group fairness requires that the proportion of individuals in a given group receiving positive (or negative) outcomes aligns with the overall proportion of those outcomes in the entire population.

The literature on group fairness presents a variety of concepts related to fairness. Each definition focuses on distinct statistical measures aimed at achieving balance among subgroups within the data. In this paper, we specifically consider demographic parity and equalized opportunity as fair metrics.

**Demographic Parity.** Demographic Parity (DP) (Zafar et al., 2017; Feldman et al., 2015; Dwork et al., 2012) mandates that the predicted outcome $\hat{Y}$ be independent of the sensitive attribute $S$, expressed mathematically as $\hat{Y} \perp S$. Most of the existing literature primarily addresses binary classification and binary attributes, where $Y \in \{0, 1\}$ and $S \in \{0, 1\}$. Similar to the concept of equal opportunity, the metric evaluating the DP fairness is defined by:

$$\triangle_{DP} = |P(\hat{Y} = 1|S = 0) - P(\hat{Y} = 1|S = 1)|. \tag{1}$$

A lower value of $\triangle_{DP}$ signifies a fairer classifier. Both Disparate Impact (DP) and Equal Opportunity (EO) metrics can be effectively extended to problems involving multi-class classifications and multiple sensitive attribute categories. This can be achieved by ensuring that $\hat{Y} \perp S$ for DP and $\hat{Y} \perp S|Y$ for EO.

**Equal Opportunity (EO).** Equal Opportunity (EO) (Hardt et al., 2016) mandates that a classifier achieves equal true positive rates across various subgroups, striving towards the ideal of a perfect classifier. The corresponding fairness measurement for EO can be articulated as follows:

$$\triangle_{EO} = |P(\hat{Y} = 1|Y = 1, S = 0) - P(\hat{Y} = 1|Y = 1, S = 1)|. \tag{2}$$

A low $\triangle_{EO}$ indicates that the difference in the probability of an instance in the positive class being assigned a positive outcome is relatively small for both subgroup members.

## 2.2 CAUCHY-SCHWARZ DIVERGENCE

Motivated by the well-known Cauchy-Schwarz (CS) inequality for square-integrable functions[1], which holds with equality if and only if $p(\mathbf{x})$ and $q(\mathbf{x})$ are linearly dependent, we can define a measure of the distance between $p(\mathbf{x})$ and $q(\mathbf{x})$. This measure is referred to as the CS divergence (Principe et al., 2000; Yu et al., 2023), given by:

$$D_{\text{CS}}(p; q) = -\log \left( \frac{\left( \int p(\mathbf{x})q(\mathbf{x})dx \right)^2}{\int p(\mathbf{x})^2 dx \int q(\mathbf{x})^2 dx} \right). \tag{3}$$

The CS divergence, denoted as $D_{\text{CS}}$, is symmetric for any two probability density functions (PDFs) $p$ and $q$, satisfying $0 \leq D_{\text{CS}} < \infty$. The minimum divergence is achieved if and only if $p(\mathbf{x}) = q(\mathbf{x})$. Given samples $\{\mathbf{x}_i^p\}_{i=1}^m$ and $\{\mathbf{x}_i^q\}_{i=1}^n$ drawn independently and identically distributed (i.i.d.) from $p(\mathbf{x})$ and $q(\mathbf{x})$ respectively, we can estimate the empirical CS divergence. This estimation can be performed using the kernel density estimator (KDE) as described in (Parzen, 1962) and follows the empirical estimator formula in (Jenssen et al., 2006).

**Proposition 1.** *Given two sets of observations $\{\mathbf{x}_i^p\}_{i=1}^{N_1}$ and $\{\mathbf{x}_j^q\}_{j=1}^{N_2}$, let $p$ and $q$ denote the distributions of two groups. The empirical estimator of the CS divergence $D_{CS}(p; q)$ is then given by:*

$$\tilde{D}_{CS}(p; q) = \log \left( \frac{1}{N_1^2} \sum_{i,j=1}^{N_1} \kappa(\mathbf{x}_i^p, \mathbf{x}_j^p) \right) + \log \left( \frac{1}{N_2^2} \sum_{i,j=1}^{N_2} \kappa(\mathbf{x}_i^q, \mathbf{x}_j^q) \right) - 2 \log \left( \frac{1}{N_1 N_2} \sum_{i=1}^{N_2} \sum_{j=1}^{N_2} \kappa(\mathbf{x}_i^p, \mathbf{x}_j^q) \right). \tag{4}$$

The proof of this proposition is detailed in Appendix A.1. where $\kappa$ represents a kernel function, such as the Gaussian kernel defined as $\kappa_\sigma(x, x') = \exp(-\|x - x'\|_2^2/2\sigma^2)$. In the following sections, we will explore the relationship between this kernel function and the existing fairness regularizer.

## 3 CAUCHY-SCHWARZ FAIRNESS REGULARIZER

In this section, we first introduce three prominent fairness regularizers that assess distribution distance using different metrics: Mean Maximum Discrepancy, Kullback-Leibler divergence, and Hilbert-

---

[1] $\left( \int p(\mathbf{x})q(\mathbf{x}) \, dx \right)^2 \leq \int p(\mathbf{x})^2 \, dx \int q(\mathbf{x})^2 \, dx$

Schmidt Independence Criterion (HSIC). For each metric, we explore its relationship with CS divergence. Subsequently, we explain how CS divergence can be utilized to achieve fairness.

### 3.1 WHAT IS THE RELATIONSHIP BETWEEN CS DIVERGENCE AND EXISTING DISTRIBUTION DISTANCE MEASURES?

To illustrate the advantages of the CS fairness regularizer, we begin by summarizing the commonly used distribution distance metrics: Maximum Mean Discrepancy (MMD), Kullback-Leibler divergence (KL), and Hilbert-Schmidt Independence Criterion (HSIC).

The definition of fairness cannot be optimized directly. Previous studies have explored various measurements of distribution distance to provide a fairness objective for optimization. Generally, the fairness objective can be summarized as follows:

$$\min_{\theta} \mathcal{L}_{utility} + \lambda \mathcal{L}_{fairness}, \tag{5}$$

where $\theta$ represents the set of model parameters that need to be learned. The term $\mathcal{L}_{utility}$ denotes the loss function that measures the utility of the model, while $\mathcal{L}_{fairness}$ indicates the fairness constraint applied in the model. The parameter $\beta$ is used to control the trade-off between utility and fairness.

**Demographic Parity Regularizer.** The demographic parity regularizer is widely utilized in fairness-focused machine learning studies (Chuang & Mroueh, 2020). It aims to optimize the mean disparity between two *prediction distributions*. This regularizer can be formally expressed as:

$$\text{DP}(p; q) = |\frac{1}{N_1} \sum_{i}^{N_1} p(\mathbf{x}_i) - \frac{1}{N_2} \sum_{j}^{N_2} q(\mathbf{x}_j)|, \tag{6}$$

where $\mathbf{x}_i$ are data points from $S = 0$, and $\mathbf{x}_j$ are data points from $S = 1$, in the context of fairness. In the following, we also represent $\mathbf{x}_i$ with distribution $p$ and $\mathbf{x}_j$ with distribution $q$ as $\mathbf{x}_i^p$ and $\mathbf{x}_i^q$ for simplicity. However, only optimizing on the mean disparity of two distributions cannot always generate an optimized DP or EO, as the Equation (6) equals 0 is a necessary but not sufficient condition for achieving DP and EO.

**Mean Maximum Discrepancy.** One of the most widely used distance metrics is the Mean Maximum Discrepancy (MMD) (Gretton et al., 2012). In the context of fairness, previous studies have employed MMD as a regularizer to enforce statistical parity among the *embeddings* of different sensitive groups within a machine learning model (Deka & Sutherland, 2023; Louizos et al., 2016). This approach aims to facilitate fair representation learning.

$$\widetilde{\text{MMD}}^2(p; q) = \frac{1}{N_1^2} \sum_{i,j=1}^{N_1} \kappa(\mathbf{x}_i^p, \mathbf{x}_j^p) + \frac{1}{N_2^2} \sum_{i,j=1}^{N_2} \kappa(\mathbf{x}_i^q, \mathbf{x}_j^q) - \frac{2}{N_1 N_2} \sum_{i=1}^{N_1} \sum_{j=1}^{N_2} \kappa(\mathbf{x}_i^p, \mathbf{x}_j^q). \tag{7}$$

By comparing with Equation (14), we observe that the CS divergence introduces a logarithmic term for each component of the MMD. Through simple transformations, we can deduce the following:

**Remark 1.** *CS divergence measures the cosine distance between $\boldsymbol{\mu}_p$ and $\boldsymbol{\mu}_q$ in a Reproducing Kernel Hilbert Space, while MMD utilizes Euclidean distance.*

**Kullback-Leibler Divergence.** Kullback-Leibler (KL) Divergence is a key concept in information bottleneck theory, where it is used to quantify the mutual information between two probability distributions. This metric has gained popularity across various domains, including fair machine learning (Kamishima et al., 2012).

$$D_{\text{KL}} = \int p(\mathbf{x}) \log \left( \frac{p(\mathbf{x})}{q(\mathbf{x})} \right) \tag{8}$$

**Hilbert-Schmidt Independence Criterion (HSIC).** Let $K$ and $L$ denote the Gram matrices for the variables $x$ and $y$, respectively. Specifically, $K$ is defined such that $K_{ij} = \kappa(\mathbf{x}_i, \mathbf{x}_j)$, and $L$ is defined as $L_{ij} = \kappa(\mathbf{y}_i, \mathbf{y}_j)$, where $\kappa$ is the Gaussian kernel function given by $\kappa = \exp\left(-\frac{\|\cdot\|^2}{2\sigma^2}\right)$. The Hilbert-Schmidt Independence Criterion (HSIC) can be estimated using the following expression (Gretton

et al., 2007):

$$\widetilde{\text{HSIC}}(p;q) = \frac{1}{N^2} \sum_{i,j}^{N} K_{ij} Q_{ij} + \frac{1}{N^4} \sum_{i,j,q,r}^{N} K_{ij} Q_{qr} - \frac{2}{N^3} \sum_{i,j,q}^{N} K_{ij} Q_{iq} = \frac{1}{N^2} \text{tr}(KHQH), \quad (9)$$

where $H = I - \frac{1}{N} \mathbb{1}\mathbb{1}^T$ represents a centering matrix of size $N \times N$. In this expression, $I$ is the identity matrix, $\mathbb{1}$ is a vector of ones, and $\frac{1}{N}\mathbb{1}\mathbb{1}^T$ computes the average across the columns, effectively centering the data by subtracting the mean from each entry.

Compared to Equation (7), The HSIC can be interpreted as the MMD between the joint distribution $p(\mathbf{x}, \mathbf{y})$ and the product of their marginal distributions $p(\mathbf{x})p(\mathbf{y})$.

### 3.2 WHY IS THE CAUCHY-SCHWARZ DIVERGENCE MORE EFFECTIVE FOR ENSURING FAIRNESS?

The Cauchy-Schwarz Divergence is particularly well-suited for promoting fairness due to several key reasons:

**(1) Closed-form solution for the mixture of Gaussians.** The CS divergence has several advantageous properties, one of which is that it provides a *closed-form solution for the mixture of Gaussians* (Kampa et al., 2011). This particular property has facilitated its successful application in various tasks, including deep clustering (Trosten et al., 2021), disentangled representation learning (Tran et al., 2022), and point-set registration (Sanchez Giraldo et al., 2017).

**(2) CS Divergence has a tighter error bound than the KL divergence.**

**Proposition 2.** *For any $d$-variate Gaussian distributions $p \sim \mathcal{N}(\boldsymbol{\mu}_p, \Sigma_p)$ and $q \sim \mathcal{N}(\boldsymbol{\mu}_q, \Sigma_q)$, where $\Sigma_p$ and $\Sigma_q$ are positive definite, the following inequality holds:*

$$D_{\text{CS}}(p;q) \leq D_{\text{KL}}(p;q) \text{ and } D_{\text{CS}}(p;q) \leq D_{\text{KL}}(q;p). \quad (10)$$

The proof can be found in Appendix A.3.

**(3) CS divergence can provide tighter bounds than MMD and DP when the distributions are far apart or when the scale of the embeddings varies significantly.** Based on the analysis presented in Remark 1, we know that CS divergence employs cosine distance, while MMD relies on Euclidean distance. In addition, DP Equation (6) utilizes a mean disparity, which is a Manhattan distance for the mean estimations of two distributions. CS divergence measures the angle between two distributions in the feature space, focusing on the difference in direction rather than magnitude. In cases where the distributions have significantly different variances or scales, MMD and DP may yield a large distance even if the distributions are aligned in the feature space. In contrast, CS divergence normalizes this comparison, resulting in a more accurate measure of similarity and thereby providing a tighter generalization bound. This normalization enhances the robustness of CS divergence, preventing MMD and DP from overestimating the discrepancy due to its reliance on an unnormalized distance measure.

### 3.3 HOW CAN THE CAUCHY-SCHWARZ DIVERGENCE BE APPLIED TO MITIGATE BIAS?

As mentioned earlier, the goal of fairness is to ensure an equal distribution of predictions across sensitive attributes. To achieve this, fairness-aware algorithms focus on minimizing the dependency of predictions on these sensitive attributes. Therefore, effectively modeling the relationship between the outcome variable $Y$ and the sensitive attribute $S$ becomes crucial. The prediction distribution over the sensitive attribute $S$ is defined as follows:

$$\mathbb{P} = P(\hat{Y} \mid S = 0); \quad \mathbb{Q} = P(\hat{Y} \mid S = 1). \quad (11)$$

By substituting the distribution of predictions over the sensitive attribute into Equation (14), where $p = \mathbb{P}$ and $q = \mathbb{Q}$, we can define the objective we aim to solve as follows:

$$\min_{\theta} \mathcal{L}_{\text{BCE}} + \alpha \tilde{D}_{\text{CS}}(\mathbb{P}, \mathbb{Q}) + \frac{\beta}{2} \|\theta\|_2^2, \quad (12)$$

where $\mathcal{L}_{\mathrm{BCE}}$ represents the binary cross-entropy loss, which measures the classifier's task-specific accuracy. It is defined as

$$\mathcal{L}_{\mathrm{BCE}} = \frac{1}{M} \sum_{i=1}^{M} -Y_i \log \hat{Y}_i,$$

where $\hat{Y}_i$ is the predicted output obtained from the training model parameterized by $\theta$. This model can be a Multi-Layer Perceptron for tabular data or a ResNet for image data. Additionally, $\|\theta\|_2^2$ serves as an $L_2$ regularizer.

## 4 EXPERIMENTS

In this section, we evaluate the effectiveness of the `CS` fairness regularizer from several perspectives: **(1)** utility and fairness performance, **(2)** the tradeoff between utility and fairness, **(3)** prediction distributions across different sensitive groups, **(4)** T-SNE plots for these sensitive groups, and **(5)** the sensitivity of parameters in Equation (12). Our evaluation encompasses five datasets with diverse sensitive attributes, including four tabular datasets: `Adult`, `COMPAS`, `ACS-I`, and `ACS-T`, as well as one image dataset, `CelebA-A`. Utility performance is assessed based on accuracy and the area under the curve (AUC), while fairness performance is measured using $\triangle_{DP}$ Equation (1) and $\triangle_{EO}$ Equation (2). Detailed information about the datasets and baselines can be found in Appendix C and Appendix D, respectively. Experimental setups are outlined in Appendix E.1, and the range for hyperparameter selection is detailed in Appendix E.2. We denote an observation drawn from the results as **Obs.**.

### 4.1 FAIRNESS AND UTILITY PERFORMANCE

We conducted experiments on five datasets along with their corresponding baselines, as previously mentioned. For each dataset, we performed 10 different splits to ensure robustness in our results. We calculated the mean and standard deviation for each metric across these splits. The accuracy and fairness performance of the downstream tasks is in Table 1. Our observations are as follows:

**Obs. 1: `CS` consistently achieves the best $\triangle_{EO}$ and ranks among the top four for $\triangle_{DP}$ across the `Adult`, `COMPAS`, and `ACS-I` datasets, with only a small margin behind the best results on the remaining datasets.** Notably, `CS` demonstrates exceptional fairness performance on the image dataset, `CelebA-A`, where the disparity in the 'Young' and 'Non-Young' groups sees a $\triangle_{DP}$ reduction of 97.36% and a $\triangle_{EO}$ reduction of 98.58%. Furthermore, in the `Adult` and `ACS-I` datasets, which include gender groups, traditional methods such as `DP`, `MMD`, `HSIC`, and `PR` do not effectively optimize for EO fairness. In contrast, the proposed `CS` achieves significant reductions in $\triangle_{EO}$ by 72.12% and 63.85%, respectively, compared to MLP.

**Obs. 2: `CS` achieves good fairness performance with a small sacrifice in utility.** Specifically, `CS` exhibits a decrease of less than 3.1% in accuracy and less than 2.2% in AUC. The only exception is observed with `COMPAS` when gender is treated as a sensitive attribute, resulting in a slightly higher accuracy loss of 3.6%. Notably, `CS` demonstrates either equivalent or improved AUC performance, with increases of 0.02% and 0.58% on `Adult` for the gender and race groups, respectively, as well as a 0.35% increase on `COMPAS` for the race group. Among the baselines, `HSIC` ranks highest in utility, achieving the best performance on `ACS-I` for the race group and on `ACS-T` for both the gender and race groups. This is followed by `PR`, which shows the best utility on `COMPAS` for both the gender and race groups, as well as on `CelebA-A` for the gender group.

### 4.2 HOW DO ACCURACY AND FAIRNESS TRADE-OFF IN BASELINE MODELS AND CS?

We evaluate the trade-off between accuracy and $\triangle_{DP}$ for the baselines by varying the fairness hyperparameters (Yao et al., 2023; Deka & Sutherland, 2023). The results are presented in Figure 2, where the x-axis represents the target accuracy, while the y-axis shows the average Demographic Parity (DP) across both positive and negative target classes. It is important to note that the figure in the bottom right corner represents the optimal result.

**Obs. 3: At the same utility level, `CS` is the most effective method in promoting fairness.** Analyzing the results, we find that `CS` consistently achieves the lowest $\triangle_{DP}$ across most accuracy

Table 1: Fairness performance of existing fair models on the tabular dataset, considering race and gender as sensitive attributes. ↑ indicates accuracy improvement compared to MLP, with higher accuracy reflecting better performance, and ↓ denotes fairness improvement relative to MLP, where lower values indicate better fairness. All results are based on 10 runs for each method. The best results for each metric and dataset are highlighted in **bold** text.

| Methods | | | Utility | | | | Fairness | | | |
|---|---|---|---|---|---|---|---|---|---|---|
| | | | ACC (%) | ↑ | AUC (%) | ↑ | $\Delta_{DP}$ (%) | ↓ | $\Delta_{EO}$ (%) | ↓ |
| Adult | Gender | MLP | 85.63±0.34 | — | 90.82±0.23 | — | 16.52±0.91 | — | 8.43±3.20 | — |
| | | DP | 82.42±0.39 | -3.75% | 86.91±0.80 | -4.31% | 1.29±0.95 | 92.19% | 20.15±1.13 | -139.03% |
| | | MMD | 81.90±0.68 | -4.36% | 85.27±0.52 | -6.11% | 2.47±0.52 | 85.05% | 17.53±1.36 | -107.95% |
| | | HSIC | 82.89±0.23 | -3.20% | 87.25±0.41 | -3.93% | 2.66±0.54 | 83.90% | 18.47±1.22 | -119.10% |
| | | PR | 81.81±0.52 | -4.46% | 85.38±0.82 | -5.99% | **0.71**±0.40 | 95.70% | 12.45±2.38 | -47.69% |
| | | CS | **83.04**±0.51 | -3.02% | **90.84**±0.35 | 0.02% | 2.13±0.89 | 87.11% | **2.35**±1.15 | 72.12% |
| | Race | MLP | 84.42±0.31 | — | 90.15±0.36 | — | 13.47±0.83 | — | 9.25±3.86 | — |
| | | DP | 83.64±0.78 | -0.92% | 88.45±0.32 | -1.89% | 2.45±0.67 | 81.81% | 2.16±1.06 | 76.65% |
| | | MMD | 83.12±0.82 | -1.54% | 88.36±0.67 | -1.99% | 2.58±0.75 | 80.85% | 3.33±0.93 | 64.00% |
| | | HSIC | **84.98**±0.17 | 0.66% | **90.90**±0.19 | 0.83% | 7.90±0.72 | 41.35% | 2.11±0.18 | 77.19% |
| | | PR | 82.13±1.16 | -2.71% | 87.44±0.33 | -3.01% | **1.53**±0.83 | 88.64% | 0.86±0.60 | 90.70% |
| | | CS | 83.14±0.86 | -1.52% | 90.67±0.22 | 0.58% | 2.76±0.56 | 79.51% | **0.47**±0.19 | 94.92% |
| COMPAS | Gender | MLP | 66.85±0.72 | — | 72.10±0.94 | — | 13.22±3.32 | — | 11.41±5.83 | — |
| | | DP | 64.20±1.58 | -3.96% | 70.64±1.05 | -2.02% | 5.78±0.33 | 56.28% | 6.78±1.61 | 40.58% |
| | | MMD | 64.82±1.62 | -3.04% | 70.72±0.92 | -1.91% | 3.09±0.92 | 76.63% | 3.15±4.37 | 72.39% |
| | | HSIC | 63.17±3.46 | -5.50% | 71.17±0.84 | -1.29% | 1.84±0.43 | 86.08% | 2.60±0.63 | 77.21% |
| | | PR | **64.95**±0.15 | -2.84% | **72.12**±0.75 | 0.03% | 3.85±0.60 | 70.88% | 3.91±1.02 | 65.73% |
| | | CS | 63.25±1.12 | -5.39% | 71.63±0.89 | -0.65% | **1.28**±0.11 | 90.32% | **0.45**±0.21 | 96.06% |
| | Race | MLP | 66.99±1.05 | — | 72.46±0.88 | — | 17.24±4.15 | — | 19.44±4.63 | — |
| | | DP | 64.98±3.72 | -3.00% | 72.09±1.03 | 0.51% | 8.70±1.12 | 49.54% | 7.04±2.13 | 63.79% |
| | | MMD | 64.41±2.04 | -3.85% | 72.10±1.83 | 0.50% | 4.42±2.11 | 74.36% | 5.60±1.25 | 71.19% |
| | | HSIC | 64.52±2.20 | -3.69% | 72.16±0.94 | 0.41% | 2.21±0.68 | 87.18% | 2.72±0.87 | 86.01% |
| | | PR | **67.22**±0.90 | 0.34% | **72.86**±0.87 | -0.55% | 5.60±1.12 | 67.52% | 6.52±1.30 | 66.46% |
| | | CS | 64.93±0.83 | -3.08% | 72.21±0.16 | 0.35% | **1.45**±0.61 | 91.59% | **1.79**±1.44 | 90.69% |
| ACS-I | Gender | MLP | 82.04±0.27 | — | 90.16±0.18 | — | 10.26±4.68 | — | 2.13±3.64 | — |
| | | DP | 81.32±0.17 | -0.88% | 89.33±0.15 | -0.92% | 0.96±0.22 | 90.64% | 5.37±0.32 | -152.11% |
| | | MMD | **80.93**±0.55 | -1.35% | 88.44±1.71 | -1.91% | 2.45±0.65 | 76.12% | 4.91±1.48 | -130.52% |
| | | HSIC | 81.40±0.12 | -0.78% | **89.53**±0.10 | -0.70% | 1.54±0.18 | 84.99% | 4.95±0.39 | -132.39% |
| | | PR | 80.03±0.30 | -2.45% | 88.10±0.26 | -2.28% | **0.35**±0.20 | 96.59% | 4.54±0.41 | -113.15% |
| | | CS | 81.72±0.75 | -0.39% | 89.24±0.42 | -1.02% | 0.86±0.41 | 91.62% | **0.77**±0.27 | 63.85% |
| | Race | MLP | 81.23±0.14 | — | 90.16±0.18 | — | 10.06±1.84 | — | 7.42±0.66 | — |
| | | DP | 81.25±0.13 | 0.02% | 89.45±0.11 | -0.79% | 0.56±0.30 | 94.43% | 4.53±0.48 | 38.95% |
| | | MMD | 80.22±1.22 | -1.24% | 88.42±1.63 | -1.93% | 1.45±0.89 | 85.59% | 4.01±0.54 | 45.96% |
| | | HSIC | **81.41**±0.15 | 0.22% | **89.67**±0.12 | -0.54% | 1.04±0.53 | 89.66% | 2.77±0.35 | 62.67% |
| | | PR | 80.27±0.26 | -1.18% | 88.45±0.21 | -1.90% | **0.37**±0.30 | 96.32% | 4.25±0.49 | 42.72% |
| | | CS | 80.15±0.68 | -1.33% | 88.23±1.01 | -2.14% | 1.02±0.58 | 89.86% | **1.94**±0.55 | 73.85% |

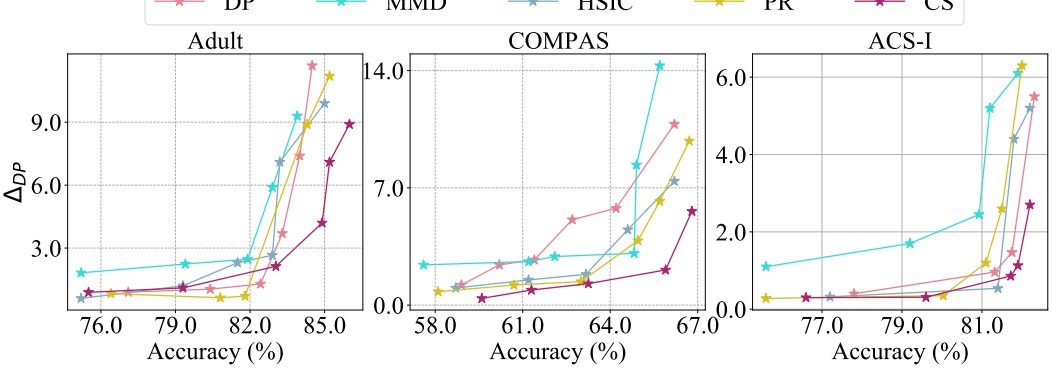

Figure 2: Fairness-accuracy trade-off curves on the test sets for (left) Adult, (middle) COMPAS, and (bottom) ACS-I. Ideally, results should be positioned in the bottom-right corner.

levels, with this effect becoming more pronounced at higher accuracy levels. This is evidenced by the significant gap in $\triangle_{DP}$ between CS and other baselines. It is important to note that while all baselines can demonstrate good fairness when the optimization prioritizes fairness over task-specific objectives (resulting in lower accuracy), the task objective remains critical for the practical application of these models. This underscores the advantage of CS, which effectively maintains the lowest bias ($\triangle_{DP}$) as accuracy improves.

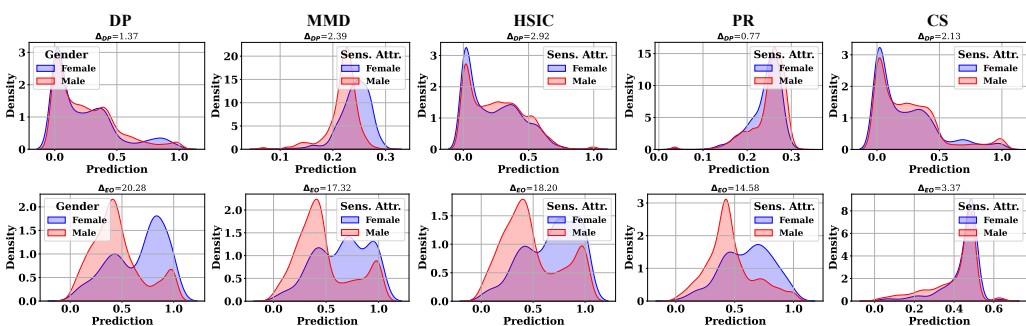

Figure 3: Prediction distributions for female and male groups in the `Adult` dataset.

**Obs. 4: High accuracy can sometimes lead to worse fairness compared to MLP, as the fairness objective becomes more challenging to optimize when there is a stronger focus on task-specific objectives.** As shown in Table 1, the $\triangle_{DP}$ for `MMD` is over 14.0, which is greater than the average $\triangle_{DP}$ of 13.22 for MLP. However, these fairness regularizers generally prove effective in controlling bias in representations, especially when more emphasis is placed on the task-specific objective. Notably, some datasets with particular sensitive attributes pose greater challenges for achieving fairness. For instance, the `COMPAS` dataset, which includes gender as a sensitive attribute, demonstrates this difficulty. One possible explanation is the relatively small sample size of `COMPAS`, which contains only $6,172$ samples—significantly fewer than other datasets where fairness is easier to achieve. For example, the `ACS-I` dataset has $195,995$ samples, approximately $31.7$ times that of `COMPAS`, and features a more balanced gender distribution.

**Obs. 5: `CS` displays a significant increase in $\triangle_{DP}$ at a slower rate than other baselines as accuracy increases.** We analyze the slope of the lines representing the increase in $\triangle_{DP}$ with rising accuracy. Many methods, such as `PR` and `DP`, demonstrate strong fairness performance at low accuracy levels; however, they quickly lose control over fairness as accuracy begins to increase. This is evident from the abrupt rise in $\triangle_{DP}$ observed at around $82.0\%$ on `Adult`, $63.0\%$ on `COMPAS`, and $81.0\%$ on `ACS-I`. In contrast, `CS` only exhibits a sudden increase at $85.0\%$, $65.5\%$, and $81.5\%$ for the same datasets, respectively. This finding further underscores the effectiveness of `CS` in maintaining a balance between utility and fairness.

### 4.3 How Can the `CS` Fairness Regularizer Improve Fairness for Both DP and EO?

We visualize the kernel density estimate plot [2] of the predictions $\hat{Y}$ across different sensitive groups to analyze how `CS` achieves a better balance of various fairness definitions compared to other baselines. The *first row* displays the predictions for all target classes, specifically $Y = 0$ and $Y = 1$, grouped by sensitive attributes. In this row, the blue areas represent the prediction density for $S = 0$, while the red areas indicate the prediction density for $S = 1$. The *second row* illustrates the prediction density for the positive target class, $Y = 1$, across two different sensitive groups. Figure 3 presents the results for `Adult` based on gender and race groups, with additional results for other datasets available in Appendix B.2.

**Obs. 6: `CS` effectively optimizes the prediction distributions for the two sensitive groups, specifically $\hat{Y}|S = 0$ and $\hat{Y}|S = 1$. Additionally, it optimizes the prediction distributions for these groups within the positive target group, i.e., $\hat{Y}|S = 0, Y = 1$ and $\hat{Y}|S = 1, Y = 1$.** Achieving DP and EO fairness requires different objectives. For instance, `DP` directly optimizes the $\triangle_{DP}$, which results in reduced effectiveness for achieving EO fairness. This is evident across all datasets, as `DP` ranks among the worst, achieving $7/10$ of the lowest EO fairness scores on $\triangle_{EO}$ when tested on five datasets with two types of sensitive attributes. The distribution plots for `DP` further illustrate this, showing a generally larger gap between the two sensitive groups in the EO plots compared to other methods. In contrast, `CS` consistently minimizes the prediction density gap between the two sensitive groups. Even in challenging cases, such as the `CelebA-A` dataset with

---

[2]https://seaborn.pydata.org/generated/seaborn.kdeplot.html

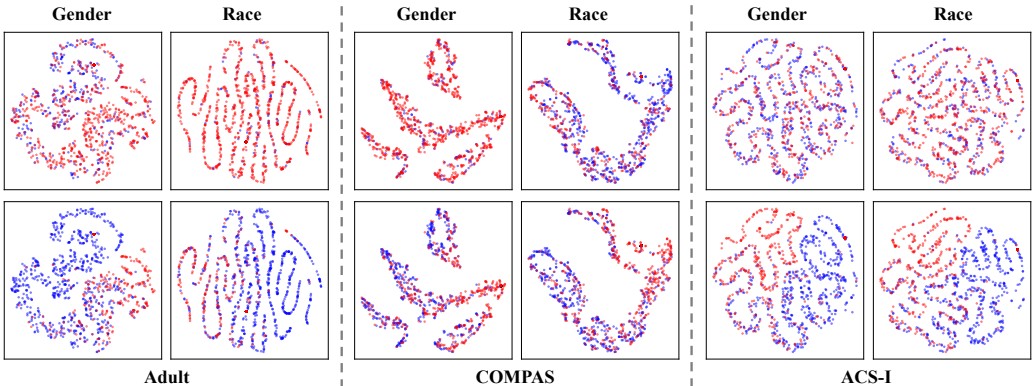

Figure 4: T-SNE visualizations of the latent representations on `Adult`, `COMPAS`, and `ACS-I`, colored by the target attribute (top) and the sensitive attribute (bottom).

gender groups, `CS` optimizes the prediction densities for female and male groups to be much closer than those of other baselines.

### 4.4 IS THE REPRESENTATION LEARNED BY APPLYING CS VIEWED AS FAIR?

To further validate that `CS` can learn fair representations, we visualize the T-SNE embeddings of the latent space from the last layer before the prediction layer (Van der Maaten & Hinton, 2008)[3]. Figure 4 displays the representations learned from the last embedding layer on the `Adult`, `COMPAS`, and `ACS-I` datasets, while Figure 11 presents the results for `ACS-T` and `CelebA-A`. Based on these visualizations, we make the following observations:

**Obs. 7: The `CS` can learn representations that are indistinguishable between sensitive groups.** This observation validates the effectiveness of `CS` in learning fair representations. Specifically, the plots in the first row of Figure 4 illustrate the embedding visualization of two sensitive groups: blue for $S=0$ and red for $S=1$. Overall, the points are uniformly dispersed, with no clear clusters of nodes sharing the same color. This indicates that the embeddings are learned independently of the sensitive attribute. Although some groups have a greater number of data points—such as in the `Adult` dataset with the sensitive attribute *race*, where the ratio of $S=0:S=1$ is $1:9.20$, and in the `COMPAS` dataset with *gender*, where the ratio is $1:4.17$ (as shown in Table 3)—the distribution of points in both colors remains even.

**Obs. 8: The `CS` can learn distinguishable representations for different target attributes.** Observing the second row of Figure 4, we can identify a distinct pattern in the distribution of the blue and red points across different locations in the plot. Among these, the embedding for `ACS-I` exhibits the clearest pattern, followed by `Adult`. This observation is consistent with the utility results presented in Table 1, which show a decrease in accuracy and AUC as the degree of negativity increases, particularly evident in the ↑ columns compared to the MLP. In contrast, `COMPAS` presents a greater challenge in ensuring utility while considering fairness, as indicated by the less distinct pattern in the learned embeddings, corroborated by the most significant utility drops in Table 1.

## 5 PARAMETER SENSITIVITY ANALYSIS

For all models, we tune the hyperparameters using cross-validation on the training set. The hyperparameters for these variants are determined through grid search during cross-validation. Specifically, we vary the parameters $\alpha$ and $\beta$ in Equation (12) across the ranges $(1e-6, 150)$ and $(1e-3, 10)$, respectively. More details regarding the hyperparameter setup and selection for all implemented methods can be found in Appendix E.1 and Appendix E.2. In this experiment, we specifically visualize the values of $\alpha$ in the range $(1e-4, 1e-1)$ for `CS`.

---

[3]https://scikit-learn.org/stable/modules/generated/sklearn.manifold.TSNE.html

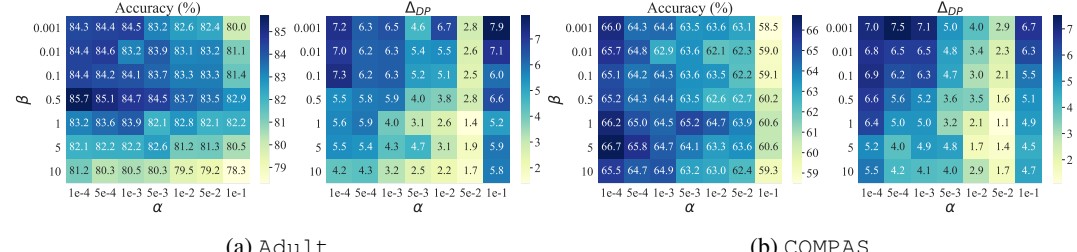

(a) `Adult`                                    (b) `COMPAS`

Figure 5: Parameter sensitivity analysis on `Adult` and `COMPAS`

The heatmap in Figure 5 illustrates the accuracy and $\triangle_{DP}$ across various combinations of $\alpha$ and $\beta$ values for the `Adult` and `COMPAS` datasets, respectively. In the accuracy plots, darker colors indicate higher values, which are preferable, while lighter colors in the $\triangle_{DP}$ plots represent better fairness performance. **Obs. 9: The highest accuracy for both `Adult` and `COMPAS` is achieved when $\alpha$ is set to its smallest value, $1e-4$, while the best fairness is obtained with $\alpha = 5e-2$.** Notably, fairness drops significantly when $\alpha$ increases from $5e-2$ to $1e-1$. Generally, smaller values of $\alpha$ can still yield satisfactory fairness performance when paired with an appropriate range of $\beta$, specifically around $5-10$. **Obs. 10: The fairness performance is more sensitive to changes in $\alpha$ than in $\beta$.** For instance, adjusting $\beta$ from $1e-3$ to $10$, which represents a $10,000\times$ increase, results in only a slight decrease in $\triangle_{DP}$ from $7.2$ to $4.2$ for `Adult`, and from $7.0$ to $5.5$ for `COMPAS`. In contrast, increasing $\alpha$ from $1e-2$ to $5e-2$, a $5\times$ change, leads to a significant drop in $\triangle_{DP}$ from $6.7$ to $2.8$ for `Adult`, and from $4.0$ to $2.9$ for `COMPAS`, when keeping $\beta$ fixed at $1e-3$.

## 6    CONCLUSION

In this paper, we introduce a novel fair machine learning method known as the Cauchy-Schwarz (CS) fairness regularizer. While existing methods effectively reduce bias in machine learning models, they often struggle to maintain balanced fairness across different fairness metrics, such as Demographic Parity (DP) and Equal Opportunity. For instance, many existing approaches can achieve very low DP but may still exhibit relatively high EO. We demonstrate that our proposed Cauchy-Schwarz fairness regularizer achieves superior and balanced fairness performance without compromising utility. This is accomplished by minimizing the Cauchy-Schwarz divergence between the prediction distribution and the sensitive attributes. Through analyzing the relationship between the CS divergence and other distance measurements, we found that the CS divergence provides a tighter bound than both the Kullback-Leibler divergence and the Maximum Mean Discrepancy, as well as the mean disparity (used in DP regularizer). This holds true particularly when the distributions are significantly different or when there is substantial variation in the scale of the embeddings. This leads to improved fairness performance in practical scenarios.

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

The appendix of this paper is structured as follows:

# Table of Contents

## A  DETAILS ON THE RELATION OF CS AND EXISTING FAIRNESS REGULARIZERS

### A.1  PROOF OF PROPOSITION 1

**Proposition 1.** *Given two sets of observations $\{\mathbf{x}_i^p\}_{i=1}^{N_1}$ and $\{\mathbf{x}q_j\}_{j=1}^{N_2}$, let $p$ and $q$ denote the distributions of two groups. The empirical estimator of the CS divergence $D_{CS}(p;q)$ is given by:*

$$\tilde{D}_{CS}(p;q) = \log\left(\frac{1}{N_1^2}\sum_{i,j=1}^{N_1}\kappa(\mathbf{x}_i^p,\mathbf{x}_j^p)\right) + \log\left(\frac{1}{N_2^2}\sum_{i,j=1}^{N_2}\kappa(\mathbf{x}_i^q,\mathbf{x}_j^q)\right) \tag{13}$$

$$- 2\log\left(\frac{1}{N_1 N_2}\sum_{i=1}^{N_1}\sum_{j=1}^{N_2}\kappa(\mathbf{x}_i^p,\mathbf{x}_j^q)\right). \tag{14}$$

*Proof.* The CS divergence is defined as:

$$D_{\mathrm{CS}}(p;q) = -\log\left(\frac{(\int p(\mathbf{x})q(\mathbf{x})\,\mathrm{d}\mathbf{x})^2}{\int p(\mathbf{x})^2\,\mathrm{d}\mathbf{x}\int q(\mathbf{x})^2\,\mathrm{d}\mathbf{x}}\right). \tag{15}$$

where $\hat{p}(\mathbf{x}) = \frac{1}{M}\sum_{i=1}^{M}\kappa_\sigma(\mathbf{x}-\mathbf{x}_j^p)$ and $\hat{q}(\mathbf{x}) = \frac{1}{N}\sum_{i=1}^{N}\kappa_\sigma(\mathbf{x}-\mathbf{x}_j^q)$ are kernel density estimation.

Then we can obtain:

$$\int \hat{p}^2(\mathbf{x})\,\mathrm{d}\mathbf{x} = \frac{1}{M^2}\sum_{i=1}^{M}\sum_{j=1}^{M}\kappa_{\sqrt{2}\sigma}(\mathbf{x}_i^p-\mathbf{x}_j^p). \tag{16}$$

By a similar approach,

$$\int \hat{q}(\mathbf{z})^2\,\mathrm{d}\mathbf{x} = \frac{1}{N^2}\sum_{i=1}^{N}\sum_{j=1}^{N}\kappa_{\sqrt{2}\sigma}(\mathbf{x}_i^q-\mathbf{x}_j^q), \tag{17}$$

and

$$\int \hat{p}(\mathbf{x})\hat{q}(\mathbf{x})\,\mathrm{d}\mathbf{x} = \frac{1}{MN}\sum_{i=1}^{M}\sum_{j=1}^{N}\kappa_{\sqrt{2}\sigma}(\mathbf{x}_i^q - \mathbf{x}_j^p). \tag{18}$$

Substituting Eqs. (16)-(18) into Eq. (15), we obtain:

$$\widetilde{D}_{\mathrm{CS}}(p;q) = \log\left(\frac{1}{M^2}\sum_{i,j=1}^{M}\kappa_{\sqrt{2}\sigma}(\mathbf{x}_i^p - \mathbf{x}_j^p)\right) + \log\left(\frac{1}{N^2}\sum_{i,j=1}^{N}\kappa_{\sqrt{2}\sigma}(\mathbf{x}_i^q - \mathbf{x}_j^q)\right) - \tag{19}$$

$$2\log\left(\frac{1}{MN}\sum_{i=1}^{M}\sum_{j=1}^{N}\kappa_{\sqrt{2}\sigma}(\mathbf{x}_i^q - \mathbf{x}_j^p)\right). \tag{20}$$

$\square$

## A.2 Proof of Remark 1

**Remark 1.** *CS divergence measures the cosine distance between $\boldsymbol{\mu}_p$ and $\boldsymbol{\mu}_q$ in a Reproducing Kernel Hilbert Space, while MMD utilizes Euclidean distance.*

*Proof.* Let $\mathcal{H}$ be a Reproducing Kernel Hilbert Space (RKHS) associated with a kernel $\kappa(\mathbf{x}_i^p, \mathbf{x}_j^q) = \langle f(\mathbf{x}_i^p), f(\mathbf{x}_j^q)\rangle_{\mathcal{H}}$ (Yu et al., 2024). The mean embeddings of two distributions $p$ and $q$ in $\mathcal{H}$ are denoted by $\boldsymbol{\mu}_p = \frac{1}{N_1}\sum_{i=1}^{N_1}f(\mathbf{x}_i^p)$ and $\boldsymbol{\mu}_q = \frac{1}{N_2}\sum_{j=1}^{N_2}f(\mathbf{x}_j^q)$ in $\mathcal{H}$, respectively. The CS divergence defined by Equation (14) can thus be written as:

$$\widetilde{D}_{\mathrm{CS}}(p;q) = -2\log\frac{\langle\boldsymbol{\mu}_p, \boldsymbol{\mu}_q\rangle_{\mathcal{H}}}{\|\boldsymbol{\mu}_p\|_{\mathcal{H}}\|\boldsymbol{\mu}_q\|_{\mathcal{H}}} = -2\log D_{\mathrm{COS}}(\boldsymbol{\mu}_p, \boldsymbol{\mu}_q)$$

Here, $\langle\cdot,\cdot\rangle_{\mathcal{H}}$ denotes the inner product in the RKHS, and $\|\cdot\|_{\mathcal{H}}$ represents the norm induced by the inner product. The mean embeddings $\boldsymbol{\mu}_p$ and $\boldsymbol{\mu}_q$ are elements of $\mathcal{H}$. Thus, the CS divergence is computed based on the cosine distance $D_{\mathrm{COS}}$ between $\boldsymbol{\mu}_p$ and $\boldsymbol{\mu}_q$.

Similarly, the Maximum Mean Discrepancy (MMD) between distributions $p$ and $q$ defined in Equation (7) can be written as:

$$\mathrm{MMD}^2(p,q) = \|\boldsymbol{\mu}_p - \boldsymbol{\mu}_q\|_{\mathcal{H}}^2 = D_{\mathrm{EUC}}(\boldsymbol{\mu}_p, \boldsymbol{\mu}_q).$$

Thus, the MMD measures the Euclidean distance between the mean embeddings of $p$ and $q$ in the RKHS $\mathcal{H}$, i.e., the $\boldsymbol{\mu}_p$ and $\boldsymbol{\mu}_q$. $\square$

## A.3 Proof of Proposition 2

**Proposition 2.** *For any $d$-variate Gaussian distributions $p \sim \mathcal{N}(\boldsymbol{\mu}_p, \Sigma_p)$ and $q \sim \mathcal{N}(\boldsymbol{\mu}_q, \Sigma_q)$ with positive definite $\Sigma_p$ and $\Sigma_q$, the following inequality holds:*

$$D_{\mathrm{CS}}(p;q) \leq D_{\mathrm{KL}}(p;q) \;\; and \;\; D_{\mathrm{CS}}(p;q) \leq D_{\mathrm{KL}}(q;p). \tag{21}$$

*Proof.* The KL divergence for $p$ and $q$ is given by:

$$D_{\mathrm{KL}}(p;q) = \frac{1}{2}\left(\mathrm{tr}(\Sigma_q^{-1}\Sigma_p) - d + (\boldsymbol{\mu}_q - \boldsymbol{\mu}_p)^\top\Sigma_q^{-1}(\boldsymbol{\mu}_q - \boldsymbol{\mu}_p) + \log\left(\frac{|\Sigma_q|}{|\Sigma_p|}\right)\right). \tag{22}$$

The CS divergence is expressed as (Kampa et al., 2011):

$$D_{\mathrm{CS}}(p;q) = -\log(d_{xy}) + \frac{1}{2}\log(d_{xx}) + \frac{1}{2}\log(d_{yy}), \tag{23}$$

$$\text{where:} \quad d_{pq} = \frac{\exp\left(-\frac{1}{2}(\boldsymbol{\mu}_p - \boldsymbol{\mu}_q)^\top(\Sigma_p + \Sigma_q)^{-1}(\boldsymbol{\mu}_p - \boldsymbol{\mu}_q)\right)}{\sqrt{(2\pi)^d|\Sigma_p + \Sigma_q|}}, \tag{24}$$

$$d_{pp} = \frac{1}{\sqrt{(2\pi)^d|2\Sigma_p|}}, \quad d_{qq} = \frac{1}{\sqrt{(2\pi)^d|2\Sigma_q|}}. \tag{25}$$

We simplify:

$$D_{\text{CS}}(p; q) = \frac{1}{2}(\boldsymbol{\mu}_q - \boldsymbol{\mu}_p)^{\top}(\Sigma_p + \Sigma_q)^{-1}(\boldsymbol{\mu}_q - \boldsymbol{\mu}_p) + \frac{1}{2}\log\left(\frac{|\Sigma_p + \Sigma_q|}{2^d\sqrt{|\Sigma_p||\Sigma_q|}}\right). \quad (26)$$

When the mean vectors differ, based on the property (Horn & Johnson, 2012), $\Sigma_q^{-1} - (\Sigma_p + \Sigma_q)^{-1}$ is positive semi-definite given $\Sigma_p = \Sigma_q$, we have:

$$2(D_{\text{CS}}(p; q) - D_{\text{KL}}(p; q)) = (\boldsymbol{\mu}_q - \boldsymbol{\mu}_p)^{\top}(\Sigma_p + \Sigma_q)^{-1}(\boldsymbol{\mu}_q - \boldsymbol{\mu}_p) - (\boldsymbol{\mu}_q - \boldsymbol{\mu}_p)^{\top}\Sigma_q^{-1}(\boldsymbol{\mu}_q - \boldsymbol{\mu}_p) \le 0. \quad (27)$$

When the covariance matrices differ, let $I$ be the $d$-dimensional identity matrix (Yin et al., 2024):

$$2(D_{\text{CS}}(p; q) - D_{\text{KL}}(p; q)) = \log\left(\frac{|\Sigma_p + \Sigma_q|}{2^d\sqrt{|\Sigma_p||\Sigma_q|}}\right) - \log\left(\frac{|\Sigma_q|}{|\Sigma_p|}\right) - \text{tr}(\Sigma_q^{-1}\Sigma_p) + d \quad (28)$$

$$= -d\log 2 + \log\left(|\Sigma_q^{-1}\Sigma_p + I|\right) + \frac{1}{2}\log\left(|\Sigma_q^{-1}\Sigma_p|\right) - \text{tr}(\Sigma_q^{-1}\Sigma_p) + d. \quad (29)$$

We have $|\Sigma_q^{-1}\Sigma_p| \le \left(\frac{1}{d}\text{tr}(\Sigma_q^{-1}\Sigma_p)\right)^d$, and $|\Sigma_q^{-1}\Sigma_p + I| \le \left(1 + \frac{1}{d}\text{tr}(\Sigma_q^{-1}\Sigma_p)\right)^d$. Thus, based on Equation (28), we can obtain:

$$2(D_{\text{CS}}(p; q) - D_{\text{KL}}(p; q)) \le -d\log 2 + d\log\left(1 + \frac{1}{d}\text{tr}(\Sigma_q^{-1}\Sigma_p)\right) \quad (30)$$

$$+ \frac{d}{2}\log\left(\frac{1}{d}\text{tr}(\Sigma_q^{-1}\Sigma_p)\right) - \text{tr}(\Sigma_q^{-1}\Sigma_p) + d. \quad (31)$$

The combined Equation (27) and Equation (31), we can obtain:

$$2(D_{\text{CS}}(p; q) - D_{\text{KL}}(p; q)) \le 0, \quad (32)$$

Similarly, we can obtain $2(D_{\text{CS}}(q; p) - D_{\text{KL}}(q; p)) \le 0$. In conclusion, we conclude:

$$D_{\text{CS}}(p; q) \le D_{\text{KL}}(p; q) \ \text{ and } \ D_{\text{CS}}(p; q) \le D_{\text{KL}}(q; p). \quad (33)$$

$\square$

## B MORE EXPERIMENTAL RESULTS

### B.1 EXPERIMENTS ON IMAGE DATASET

In this section, we present the experimental results on the `CelebA-A` image dataset. The `CelebA-A` face attributes dataset (Liu et al., 2015) contains over $200,000$ face images, where each image has $40$ human-labeled attributes. Among the attributes, we select 'Attractive' as a binary classification task and consider 'Gender' and 'Young' as sensitive attributes. The results are presented in Table 2.

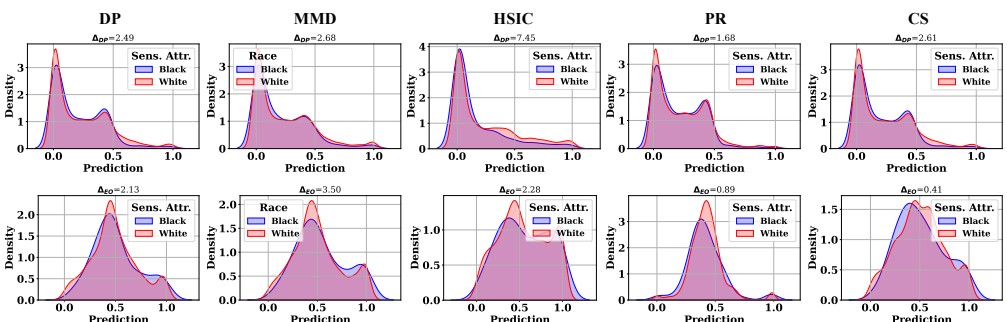

Figure 6: Prediction distributions for black and white groups in the `Adult` dataset.

Table 2: The fairness performance on the tabular dataset for existing fair models and we consider race and gender as sensitive attributes. A higher accuracy metric indicates better performance. ↑ represents the accuracy improvement compared to MLP. A lower fairness metric indicates better fairness. ↓ represents the improvement of fairness compared to MLP. The results are based on 10 runs for all methods.

|  | Methods | | Utility | | | | Fairness | | | |
|---|---|---|---|---|---|---|---|---|---|---|
|  |  |  | ACC (%) | ↑ | AUC (%) | ↑ | $\Delta_{DP}$ (%) | ↓ | $\Delta_{EO}$ (%) | ↓ |
| ACS-T | | MLP | 66.21±0.95 | — | 73.78±0.25 | — | 8.32±2.67 | — | 5.11±3.55 | — |
| | Gender | DP | 65.38±0.29 | -1.25% | 72.40±0.38 | -1.87% | 0.29±0.15 | 96.51% | 1.83±0.26 | 64.19% |
| | | MMD | 64.48±0.27 | -2.61% | 72.92±0.31 | -1.17% | 1.22±0.36 | 85.34% | 2.11±0.49 | 58.71% |
| | | HSIC | **66.01**±0.29 | -0.30% | **73.16**±0.32 | -0.84% | 0.98±0.26 | 88.22% | 1.00±0.28 | 80.43% |
| | | PR | 62.72±1.01 | -5.27% | 69.36±0.85 | -5.99% | 0.78±0.50 | 90.63% | 1.07±0.36 | 79.06% |
| | | **CS** | 65.95±0.70 | -0.39% | 72.29±0.92 | -2.02% | **0.18**±0.13 | 97.84% | **0.92**±0.63 | 82.00% |
| | | MLP | 66.38±0.42 | — | 73.69±0.63 | — | 9.28±1.63 | — | 6.21±1.63 | — |
| | Race | DP | 64.96±0.23 | -2.14% | 71.86±0.23 | -2.48% | 0.82±0.33 | 91.16% | 1.30±0.26 | 79.07% |
| | | MMD | 65.71±0.65 | -1.01% | 70.57±0.52 | -4.23% | 3.97±0.97 | 57.22% | 1.55±0.79 | 75.04% |
| | | HSIC | **65.81**±0.24 | -0.86% | **72.92**±0.23 | -1.04% | 1.75±0.31 | 81.14% | **0.43**±0.23 | 93.08% |
| | | PR | 64.25±0.87 | -3.21% | 70.25±0.30 | -4.67% | 1.56±0.87 | 83.19% | 1.21±0.74 | 80.52% |
| | | **CS** | 65.29±0.58 | -1.64% | 72.18±0.69 | -2.05% | **0.43**±0.27 | 95.37% | 1.32±0.27 | 78.74% |
| CelebA-A | | RN | 78.14±0.47 | — | 86.58±0.55 | — | 51.66±0.97 | — | 35.67±1.11 | — |
| | Gender | DP | 62.42±4.79 | -20.12% | 66.86±3.19 | -22.78% | **0.46**±0.25 | 99.11% | 4.84±2.37 | 86.43% |
| | | MMD | 62.54±4.26 | -19.96% | 66.47±3.85 | -23.23% | 1.39±0.64 | 97.31% | 5.89±3.12 | 83.49% |
| | | HSIC | 63.39±3.63 | -18.88% | 69.33±3.25 | -19.92% | 2.24±0.36 | 95.66% | 3.83±2.22 | 89.26% |
| | | PR | **65.51**±3.52 | -16.16% | **71.70**±2.88 | -17.19% | 4.00±0.52 | 92.26% | 5.05±2.57 | 85.84% |
| | | CS | 64.36±4.52 | -17.64% | 70.22±3.57 | -18.90% | 0.82±0.34 | 98.41% | **1.12**±1.14 | 96.86% |
| | | RN | 78.14±0.47 | — | 86.67±0.53 | — | 41.74±1.17 | — | 18.35±1.56 | — |
| | Young | DP | 66.78±3.61 | -14.54% | **73.95**±3.44 | -14.68% | 2.43±0.83 | 94.18% | 0.91±1.77 | 95.04% |
| | | MMD | **65.82**±4.87 | -15.77% | 72.84±3.61 | -15.96% | 3.49±0.83 | 91.64% | 1.60±0.71 | 91.28% |
| | | HSIC | 66.04±3.01 | -15.49% | 73.08±2.69 | -15.68% | 1.99±0.55 | 95.23% | 1.04±0.60 | 94.33% |
| | | PR | 62.98±4.69 | -19.40% | 69.63±4.02 | -19.66% | 1.32±0.49 | 96.84% | 1.82±0.53 | 90.08% |
| | | CS | 65.63±3.51 | -16.01% | 72.14±3.84 | -16.76% | **1.10**±0.37 | 97.36% | **0.26**±0.63 | 98.58% |

The results show a similar finding with the tabular dataset, demonstrating that 1) `DP` method always achieves a lower $\Delta_{DP}$ but a relatively high $\Delta_{EO}$. 2) `HSIC` is a more promising fair model to achieve equal opportunity.

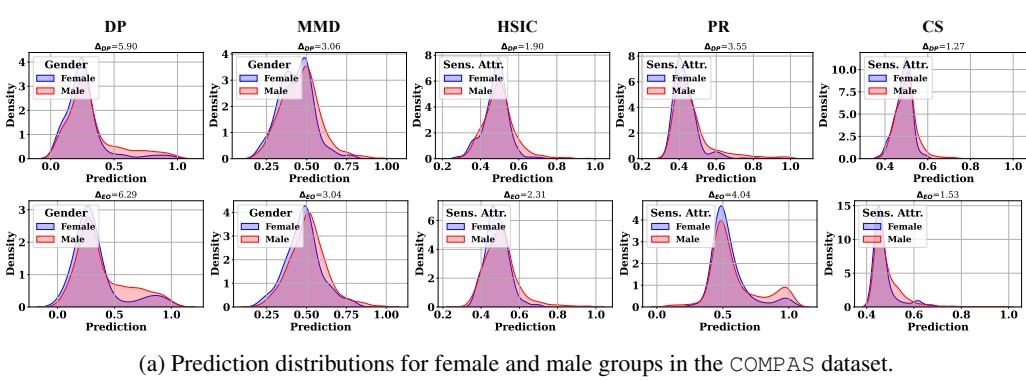

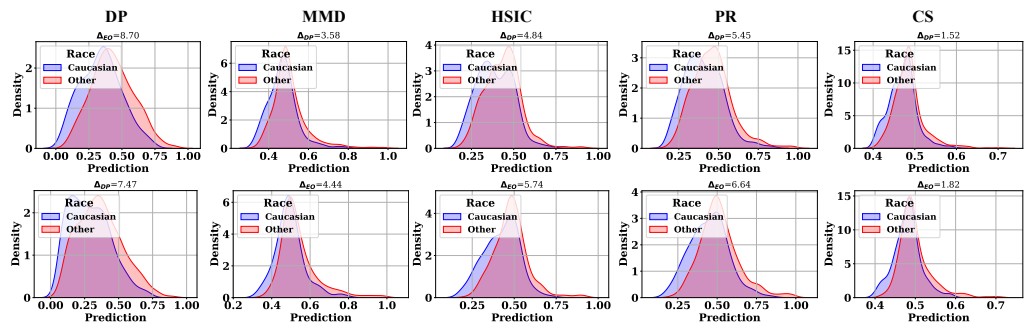

(b) Prediction distributions for Caucasian and (all) other groups in the COMPAS dataset.

Figure 7: Accuracy and $\triangle_{DP}$ trade-off on COMPAS with sensitive attribute gender and race. Results located in the bottom-right corner are preferable.

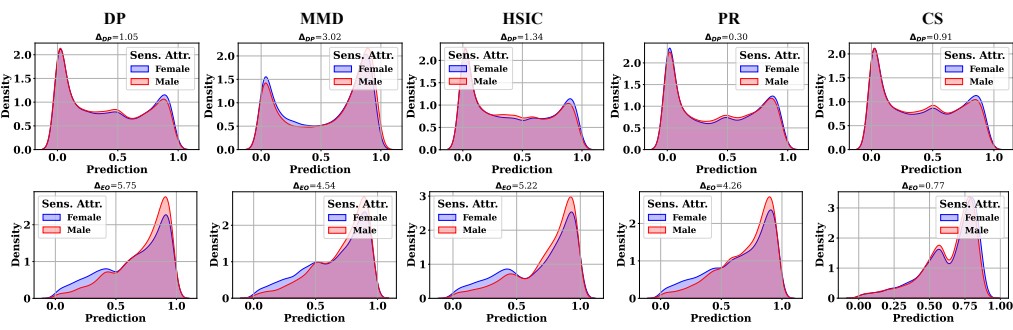

(a) Prediction distributions for female and male groups in the ACS-I dataset.

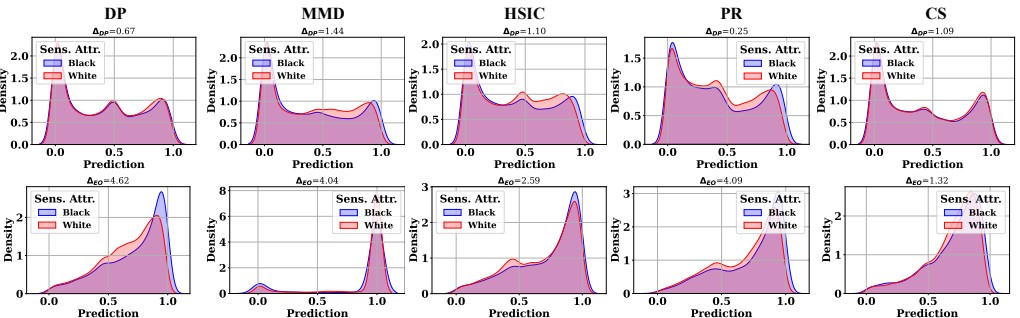

(b) Prediction distributions for black and white groups in the ACS-I dataset.

Figure 8: Accuracy and $\triangle_{DP}$ trade-off on ACS-I with sensitive attribute gender and race. Results located in the bottom-right corner are preferable.

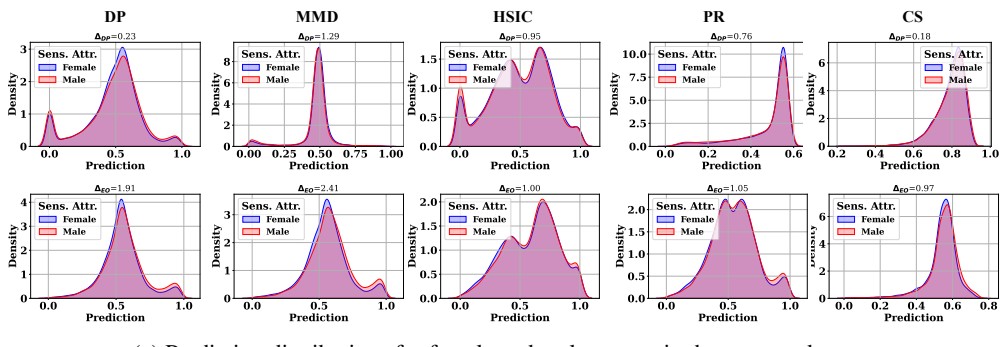

(a) Prediction distributions for female and male groups in the `ACS-T` dataset.

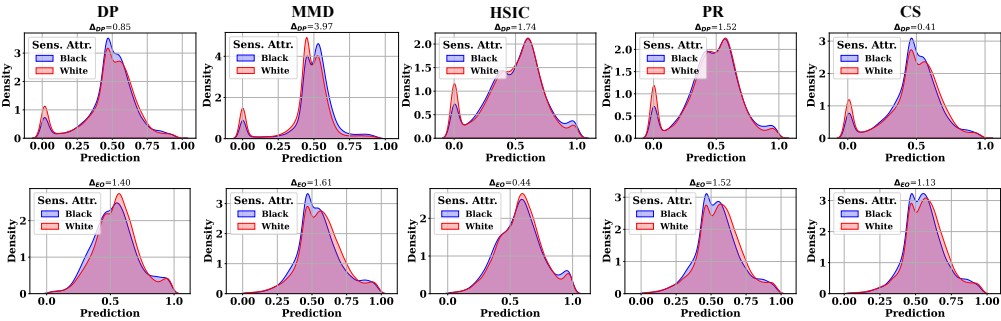

(b) Prediction distributions for black and white groups in the `ACS-T` dataset.

Figure 9: Accuracy and $\triangle_{DP}$ trade-off on `ACS-T` with sensitive attribute gender and race. Results located in the bottom-right corner are preferable.

### B.2 MORE PREDICTION DISTRIBUTIONS OVER THE SENSITIVE GROUPS

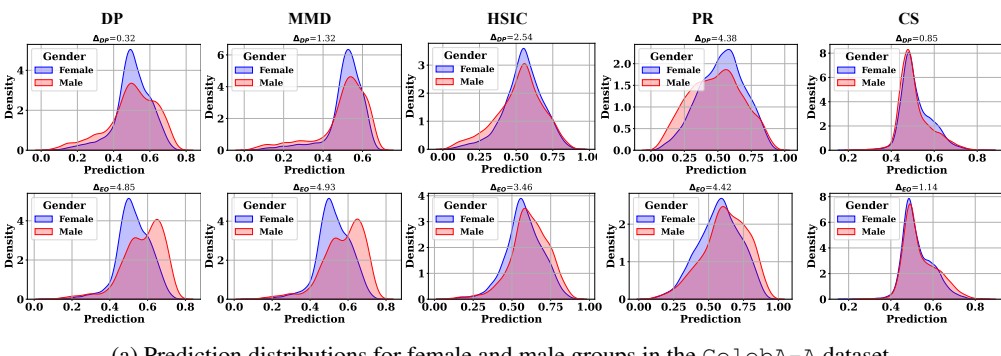

(a) Prediction distributions for female and male groups in the `CelebA-A` dataset.

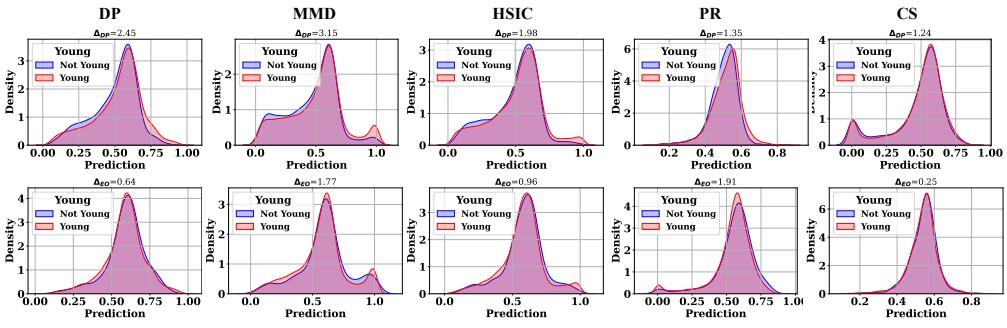

(b) Prediction distributions for young and non-yong groups in the `CelebA-A` dataset.

Figure 10: Accuracy and $\triangle_{DP}$ trade-off on `CelebA-A` with sensitive attribute gender and race. Results located in the bottom-right corner are preferable.

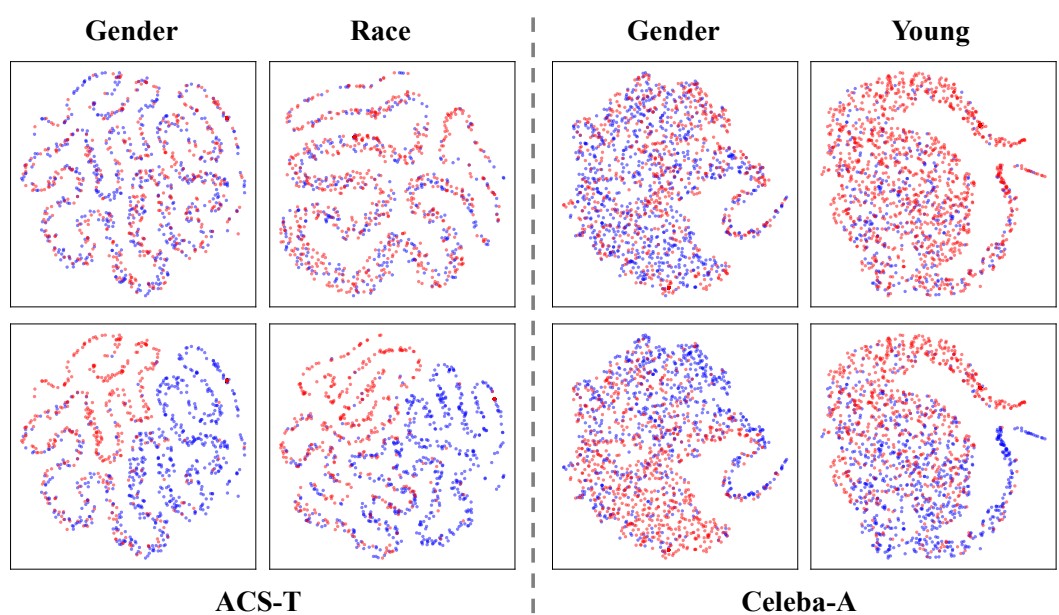

Figure 11: Accuracy and $\triangle_{DP}$ trade-off on `ACS-T` and `CelebA-A`. Results located in the bottom-right corner are preferable.

### B.3 MORE T-SNE PLOTS

In addition to the T-SNE plots shown in Figure 4 which shows the results on three datasets, we also include the T-SNE plots on two remaining datasets `ACS-T` and `CelebA-A` in Figure 11.

## C DATASET DESCRIPTIONS AND DETAILS

We conducted experiments on five datasets, including four tabular datasets and one image data. The introduction of these datasets is as below:

- **Adult**[4] (Dua & Graff, 2017) The `Adult` dataset contains information on $45,222$ individuals from the 1994 US Census. The task is to predict whether an individual's income exceeds \$50k USD based on various personal attributes. In our analysis, we consider gender and race as sensitive attributes.
- **COMPAS**[5] (Larson et al., 2016). The `COMPAS` dataset consists of records of criminal defendants and is used to predict the likelihood of recidivism within two years. The dataset includes various attributes related to the defendants, such as their criminal history, as well as demographic information, including gender and race.
- **ACS-I** and **ACS-T**[6] (Ding et al., 2021). The ACS dataset is derived from the American Community Survey (ACS) Public Use Microdata Sample and includes several prediction tasks. These tasks involve predicting attributes such as whether an individual's income exceeds \$50k or whether an individual is employed. Each task includes features such as race, gender, and other relevant characteristics specific to the task.
- **CelebA-A**[7] (Liu et al., 2015) The CelebFaces Attributes dataset contains $20,000$ face images of $10,000$ different celebrities. Each image is annotated with $40$ binary labels that represent various facial attributes, such as gender, hair color, and age. In this study, we choose 'attractive' as the target label and perform a binary classification task, while considering 'young' and 'gender' as sensitive attributes.

---

[4]https://archive.ics.uci.edu/ml/datasets/adult
[5]https://github.com/propublica/compas-analysis
[6]https://github.com/zykls/folktables
[7]https://mmlab.ie.cuhk.edu.hk/projects/CelebA.html

The detailed statistics for the aforementioned datasets are summarized as follows:

Table 3: The table presents the statistics of the datasets. #Feat. refers to the total number of features after preprocessing [8]. The ratio $0:1$ represents the proportion between the two categories of the target label or sensitive attributes.

| Dataset | Task | Sen. Attr. ($S$) | #Samples | #Feat. | Class $Y$ $0:1$ | 1st $S$ $0:1$ | 2nd $S$ $0:1$ |
|---------|------|------------------|----------|--------|-----------------|---------------|---------------|
| Adult | Income | Gender, Race | $45,222$ | $101$ | $1:0.33$ | $1:2.08$ | $1:9.20$ |
| COMPAS | Credit | Gender, Race | $6,172$ | $405$ | $1:0.83$ | $1:4.17$ | $1:0.52$ |
| ACS-I | Income | Gender, Race | $195,665$ | $908$ | $1:0.70$ | $1:0.89$ | $1:1.62$ |
| ACS-T | Travel Time | Gender, Race | $172,508$ | $1,567$ | $1:0.94$ | $1:0.89$ | $1:1.61$ |
| CelebA-A | Attractive | Gender, Young | $202,599$ | $48 \times 48$ | $1:0.95$ | $1:0.71$ | $1:3.45$ |

## D    BASELINES DETAILS

We consider four widely used fairness methods: DP, MMD, HSIC, and PR. Specifically, DP and HSIC minimize the demographic parity and Hilbert-Schmidt Independence Criterion, correspondingly. MMD learns a classifier that optimizes the Mean Maximum Discrepancy. PR optimizes the Kullback-Leibler divergence. We also include base models MLP and RN for tabular data and image data, correspondingly.

- DP: It is a gap regularization method for demographic parity (Chuang & Mroueh, 2020). As these fairness definitions cannot be optimized directly, gap regularization is differentiable and can be optimized using gradient descent.
- MMD: The Maximum Mean Discrepancy (MMD) (Gretton et al., 2012) is a metric used to measure the distance between probability distributions. Previous research has leveraged MMD to enhance fairness in machine learning models, specifically in variational autoencoders (Louizos et al., 2016) and MLPs (Deka & Sutherland, 2023). In this paper, we build on the methodologies from earlier works (Zhao & Meng, 2015) to compute the MMD baseline.
- HSIC: It minimizes the Hilbert-Schmidt Independence Criterion between the prediction accuracy and the sensitive attributes (Gretton et al., 2005; Baharlouei et al., 2020; Li et al., 2019).
- Prejudice Remover (PR) (Kamishima et al., 2012) (Prejudice Remover) minimizes the mutual information between the prediction accuracy and the sensitive attributes.

## E    MORE EXPERIMENTAL DETAILS

In this section, we describe the details of the experimental setup. In this work, we adopted a straightforward stopping strategy. We employ a linear decay strategy for the learning rate, halving it every 50 training step. The model training is stopped when the learning rate decreases to a value below $1e^{-5}$. Across all datasets, we use a weight decay of $0.0$, StepLR with a step size of $50$ and a gamma value of $0.1$, and train for $150$ epochs using the Adam Optimizer (Kingma & Ba, 2014). The batch size and learning rate vary depending on the dataset, with specific values provided below. Additionally, Table 4 lists the range of the control hyperparameter $\lambda$ for each fairness approach.

### E.1    HYPERPARAMETER SETTINGS

**1. Training Hyperparameters:**

- Tabular data (Adult, COMPAS, ACS-I, and ACS-T):
    - Learning rate: $1e^{-2}$
    - Weight decay: $0.0$
    - StepLR_step: $50$

---

[8]We adopt the preprocessing in previous studies (Le Quy et al., 2022; Mehrabi et al., 2021) involving identifying the target labels and sensitive attributes, and then selecting the relevant features for the analysis.

  - – StepLR_gamma: 0.1
  - – Training epochs: 150
  - – Batch sizes: $1,024$ on `Adult`, $32$ on `COMPAS`, $4,096$ on `ACS-I`, $4,096$ on `ACS-T`
- Image data (`CelebA-A`):
  - – Learning rate: $1e^{-3}$
  - – Weight decay: $0.0$
  - – StepLR_step: 50
  - – StepLR_gamma: 0.1
  - – Training epochs: 150
  - – Batch sizes: 256.

**2. Architecture Hyperparameters:**

- Multilayer perceptron:
  - – Number of layers: 3
  - – Number of hidden neurons: $\{512, 256, 64\}$
- ResNet-18 (He et al., 2016):
  - – Model: https://github.com/pytorch/vision/blob/main/torchvision/models/resnet.py

### E.2 HYPERPARAMETER SELECTION

To implement `CS` and the baseline methods, we adjust the hyperparameter $\lambda$ by tuning it within a specified range. The details of the hyperparameter selection process and the specific range for $\lambda$ are provided below:

Table 4: The selections of fairness control hyperparameter $\lambda$.

| Method | Fairness Control Hyperparameter $\lambda$ |
|---|---|
| DP | $0.5, 1.0, 1.2, 1.4, 1.6, 1.8, 2.0, 2.5, 3.0, 3.5, 4$ |
| HSIC | $0.1, 1, 5, 10, 50, 100, 200, 300, 400, 500, 600, 700, 800, 900, 1,000$ |
| PR | $0.05, 0.2, 0.3, 0.40, 0.50, 0.7, 0.9, 1.0$ |
| ADV | $0.5, 1.0, 1.2, 1.4, 1.6, 1.8, 2.0, 2.5, 3.0, 3.5$ |
| CS | $1e^{-6}, 1e^{-5}, 1e^{-4}, 1e^{-3}, 1e^{-2}, 2e^{-2}, 5e^{-2}, 0.1, 0.5, 1.0, 2.0, 3.0, 4.0, 50, 150$ |

## F RELATED WORK

In this section, we first review relevant prior studies, beginning with an overview of algorithmic fairness in machine learning. We then narrow our focus to regularization-based in-processing methods, which are central to our approach.

### F.1 ALGORITHMIC FAIRNESS IN MACHINE LEARNING

The importance of fairness in machine learning has grown significantly as the demand for unbiased decision-making models for individuals and groups increases. This is especially critical in high-stakes applications where the consequences of biased decisions can be severe. Fairness is commonly categorized into three main types: *Individual fairness* (Yurochkin et al., 2019; Mukherjee et al., 2020; Yurochkin & Sun, 2020; Kang et al., 2020; Mukherjee et al., 2022), which aims to ensure that similar individuals are treated similarly; *Group fairness* (Hardt et al., 2016; Verma & Rubin, 2018; Li et al., 2020; Ling et al., 2023), which focuses on achieving fairness across predefined subgroups, often defined by sensitive attributes such as gender or race; *Counterfactual fairness* (Kusner et al., 2017; Agarwal et al., 2021; Zuo et al., 2022), which seeks to ensure fairness by considering how decisions would hold under alternative scenarios. Given the widespread adoption of group fairness metrics in real-world applications and the increasing development of in-processing techniques for

deep neural network models, we focus on benchmarking these methods to ensure group fairness in neural networks, particularly for tabular and image data.

Various techniques for mitigating bias in machine learning models can be categorized into three main approaches: *pre-processing*, *in-processing*, and *post-processing*. *Pre-processing* methods focus on addressing biases present in the dataset itself to ensure that the trained model exhibits fairness (Kamiran & Calders, 2012; Calmon et al., 2017a). For instance, these techniques may involve rebalancing the dataset or modifying the data collection process (Calmon et al., 2017b). *In-processing* methods, on the other hand, adjust the training objectives by incorporating fairness constraints directly into the learning process (Kamishima et al., 2012; Zhang et al., 2018; Madras et al., 2018; Zhang et al., 2022; Buyl & De Bie, 2022; Alghamdi et al., 2022; Shui et al., 2022; Mehrotra & Vishnoi, 2022). This approach aims to ensure that the model learns fair representations during training. Finally, *post-processing* methods modify the predictions made by classifiers after the model has been trained, with the goal of promoting fairness across different groups (Hardt et al., 2016; Jiang et al., 2020; Tsaousis & Alghamdi, 2022). By categorizing these techniques, we can better understand the different strategies available for mitigating bias in machine learning systems.

## F.2 REGULARIZATION-BASED IN-PROCESSING METHODS

In this paper, we explore three types of regularization-based in-processing methods. First, *Gap Regularization* Chuang & Mroueh (2020) streamlines the optimization process by offering a smooth approximation of real-world loss functions, which are typically non-convex and difficult to optimize directly. This category includes methods such as `DP`, `EO`, and `EOD`. Second, the *Independence* approach integrates fairness constraints into the optimization, aiming to mitigate the influence of protected attributes on model predictions while maintaining overall performance. Notable examples of this approach include `PR` (Kamishima et al., 2012) and `HSIC` (Li et al., 2019). Lastly, *adversarial debiasing* seeks to minimize utility loss while hindering an adversary's ability to accurately predict the protected attributes. This approach encompasses methods like ADV (Zhang et al., 2018; Louppe et al., 2017; Beutel et al., 2017; Edwards & Storkey, 2015; Adel et al., 2019) and LAFTR (Madras et al., 2018).

