# OpenReview forum: "Cauchy-Schwarz Fairness Regularizer"
_ICLR.cc/2025/Conference — Submitted to ICLR 2025_

### Official Review · Reviewer_yoVn · 2024-10-31

**Soundness:** 1
**Presentation:** 2
**Contribution:** 1
**Rating:** 3
**Confidence:** 4

**Summary:**

This paper presents an in-processing fairness methodology, a *Cauchy-Schwarz Fairness Regularizer.* The authors consider two popular group fairness metrics in the binary classification setting: *Statistical Parity* and *Equalized Odds.* The proposed method optimizes for standard classification accuracy (through binary cross-entropy loss) while tacking on a regularizer that measures the "Cauchy-Schwarz Divergence" between two distributions: $\mathrm{Pr}(\hat{Y} \mid S = 0)$ and $\mathrm{Pr}(\hat{Y} \mid S = 1)$, the distributions of the classifier conditional on the sensitive attribute applying or not applying ($\hat{Y} \in \{0, 1\}$ is the output of the classifier, supposedly on a given example, and $S \in \{0, 1\}$ is the sensitive attribute). The authors claim that the Cauchy-Schwarz divergence give a couple of intuitive explanations for why this regularizer might be better than existing regularizers based on known divergence measures such as KL divergence, maximum mean discrepancy, and simply regularizing based on mean demographic parity rates. Then, they conduct a suite of experiments and show that the Cauchy-Schwarz Fairness Regularizer consistently achieves good simultaneous Equalized Odds and Demographic Parity, with small sacrifices to accuracy, compared to the other baseline fairness regularizers based on discrepancy measures.

**Strengths:**

**Originality:** The paper does present a new (to my knowledge) idea in using a Cauchy-Schwarz divergence to regularize their fairness objective. However, I believe that the extent of the original contribution is the use of a different divergence metric to simply regularize binary cross-entropy loss, providing yet another in-processing fairness method based on a simple and standard modification to incentivize the classifier's prediction distributions to be similar across sensitive attributes. The authors also don't provide much past a couple of surface-level remarks comparing their choice of discrepancy ("Cauchy-Schwarz divergence") to other standard divergences. For example, it is unclear whether *Proposition 2* even means anything salient in the context of this study. The authors claim that the CS divergence has "tighter error bound" than KL divergence and proceed to justify this (without any further explanation) by referencing the fact that the CS divergence between two Gaussians is always smaller than the KL divergence between two Gaussians. It is unclear to me why this would be an attractive property past the fact that the units representing the divergence between two distributions is simply smaller than the units with KL divergence, as the authors use the CS divergence to measure discrepancy between the two distributions directly. More on this in *Weaknesses.*

**Quality:** The authors present a comprehensive suite of experiments on popular fairness datasets. The Experiments section is relatively well-organized, and the results seem significant, at least in relation to the chosen baselines (although it isn't clear to me that these are the right baselines, see *Weaknesses*). The quality of the experimental results, overall, seem promising when taken at face value, but it is unclear to me whether the presented baselines are sufficient.

**Clarity:** The paper's presentation is generally clear and well-written, at least in organization. The sections flow in a readable way, and the authors take care in presenting their results in an interpretable manner. However, (see *Weaknesses*) there are numerous typos that overall obscure the clarity in the more technical statements, particularly in Section 3. Of particular note is Section 3.2, where the authors present some relatively hand-wavy "reasons" why the CS divergence measure might be a better choice than other divergence measures. In these sections, I found that the explanations from the authors were often lacking in rigor or depth. The Experiments section and the accompanying Appendix for the experimental details, however, seemed well-written and clear, at the least.

**Significance:** As stated above, I don't find this technique particularly novel past the use of simply adjusting a measure of distance in the regularizer for a standard binary classification objective. To make this a stronger paper, I believe that Section 3 should be substantially stronger, and sharper proofs and theorems should be given to compare the CS divergence measure to existing divergence measure, and, perhaps, prove some theorems that state its properties either in-sample or out-of-sample. However, these details are missing from the current state of the paper.

**Weaknesses:**

I have pointed out several weaknesses in the *Strengths* section. Overall, I don't believe this paper is in the state to be published, for several reasons. I believe there are serious limitations to consider and revisions to be made to the paper, and the it is unclear to me whether the main result is at all significant. To elaborate further:

**Typographical or clarity confusion.** Although the paper is overall clear and well-written, there are numerous flaws in the presentation of many of the technical statements in the paper. I would suggest in some of the more technical portions:

- In Equations 1 and 2, it should be mentioned that these notions maintain *in expectation* over all of the input space. That is, $P(\hat{Y} = 1 \mid S = 0)$, for example, holds in expectation over all $X$. In fact, it is slightly unclear that your random variable $\hat{Y}$ subsumes the dependence on $X$. I would either add that to the notation as well, to indicate dependence on the input. This is especially important in your later exposition, when you define probability densities over the input space that depend on the predictions $\hat{Y}$.
- On that above point, it might be helpful in general to have a separate "Notations" section to get this stuff straightened out.
- Equation 5 should have the two losses depend on the parameter $\theta$. This is a nitpick, but writing a minimization problem over a parameter when your functions don't depend on a parameter isn't quite clear.
- Nitpick: You say, "the parameter $\beta$" when it should (I think) be $\lambda$.
- Equation 6: The notation here is unclear --- you say that $p$ and $q$ are densities, so $p(\mathbf{x}_i)$ and $q(\mathbf{x}_j)$. Densities are zero on points, so $DP(p;q)$ is always zero. Perhaps you mean to write this with indicators on $\mathbf{x}_i^p$ and $\mathbf{x}_i^q$?
- Remark 1: While the proof in the Appendix seems correct, this seems to be a bit of a non sequitur. How are we meant to interpret this random factoid? Plus, $\mathbf{mu}_p$ and $\mathbf{mu}_q$ are undefined in this remark.
- In the HSIC, $L$ is defined and never used; perhaps in Equation 9 you mean to use $L$ instead of $Q$?
- Equation 11: I believe this would be much clearer if you introduced the dependence on $\mathbf{x}$. Written in Equation 11, $\mathbb{P}$ and $\mathbb{Q}$ seem to be just numbers (you present these distributions as average rates over for the classifier over $S = 0$ and $S = 1$).
- Equation 12: the dependence on $\theta$ should be made explicit (every term should depend on $\theta$).
- Table 1 and Experimental Results: The definition of each baseline should be included in the main body, not the appendix.

**Section 3: Insufficient rigor in justifying the CS divergence.** Overall, I found Section 3 seriously lacking in rigor or depth, particularly in the presentation of the "three key reasons" in Section 3.2. For the first reason, it is unclear without a rigorous statement what the authors mean by "closed-form solution for the mixture of Gaussians." The authors just state this without providing further rigorous justification or explanation, and then they simply cite a couple of additional references without any further justification of why this might be particularly helpful in this application.  For the second reason, it is unclear why the divergence from CS being smaller than KL provides any particular advantage at all, as the authors are just using it as a regularizer. Why would this scale matter at all in the regularization? What do the authors mean by "tighter error bound?" Why does the justification with Gaussian data have any bearing on the experimental results? These questions all should be answered in a revision, as they are unclear to me. The third reason is doesn't give any references and provides a non-rigorous, vague explanation comparing CS divergence to MMD and DP as divergences. I found this explanation particularly "hand-wavy," as it relies on no mathematical justification or experimental backing. Overall, the explanation behind the mechanism of the CS regularizer is unclear, and I did not gain any insight from its comparison to other divergence measures in Section 3. I believe that, if the CS divergence really is a worthwhile object of study, there should be much more rigorous justification of its properties in Section 3.

**Section 4: Insufficient suite of experiments.** Although it makes sense to compare to other inprocessing techniques for fairness, it would be helpful to also compare these baselines to other fairness techniques at other stages of the pipeline, particularly preprocessing and postprocessing. In particular, I believe that this would provide a more comprehensive suite of experimental results especially because the authors make no theoretical connections between the objective and the out-of-sample performance.

**Questions:**

The main question I am still confused above from this work that was not addressed was how these results are compatible with the classic "Impossibility Theorem" of Fair ML. In particular, it is well-known (see, e.g. Kleinberg, Mullainathan, Raghavan 2016) that it is impossible to achieve simultaneous Statistical Parity and Equalized Odds under relatively mild conditions on the binary classification problem. However, this is exactly what the authors of this paper seek to achieve with the CS regularizer, and one of their main stated contributions is a *simultaneous* control of both. However, this classic impossibility result is not cited in the paper. In light of this result, it seems to me that another plausible explanation of the experimental results isn't that it can "improve fairness for both DP and EO" as Section 4.3 suggests, but, rather, that it is just a good regularizer for EO proper.

My main question is: how does the CS regularizer stand in light of the impossibility result? Of course, I believe that the impossibility result applies to DP = 0 and EO = 0, but I believe that it throws one of the contributions and motivations of this paper into question.

Along with this question, several others (also raised in the above sections) are:
- How does this method fare in comparison to preprocessing and postprocessing methods?
- Are there more rigorous justifications for the CS regularizer than the ones mentioned in Section 3?

---

> ### Author Response · Authors · 2024-11-28
> **Response to Reviewer yoVn (Part 1/2)**
>
> We thank Reviewer yoVn for the time and efforts in review our paper. To address the reviewer's questions, we provide detailed responses below.
>
>
> **[Equation 1 hold in expectation over all X.]**
> Thank you for your suggestion. In our current paper, we introduced $X$ prior to presenting the fairness metrics.
>
> **[Fairness Coefficient.]** Thank you for pointing this out. To clarify, the coefficient $\lambda$ is introduced to represent fairness in general frameworks and baseline methods, while $\alpha$ and $\beta$ are specifically used as fairness coefficients in our proposed framework.
>
>
> **[Equation 6.]**
>
> Thank you for pointing this out. You are correct that Equation 6 should still involve ${\bf x}_i^p$ and ${\bf x}_i^p$. The simplification is introduced only after Equation 6 for clarity. Specifically, as stated in the paper:
>
> *In the following, we represent ${\bf x}_i$ with distribution $p$ and ${\bf x}_j$ with distribution $q$ as ${\bf x}_i^p$ and ${\bf x}_j^q$, respectively, for simplicity.*
>
> We have addressed this issue in our paper. Thank you for bringing this to our attention.
>
> **[Undefined ${\mu}_p$ and ${\mu}_q$.]**
>
> Thank you for the question. To improve clarity and logical flow, we have moved the definitions of $\mu_p$ and $\mu_q$ from the proof of Remark 1 to the section where they are first introduced in Remark 1. The empirical mean embeddings are defined as: $\mu\_p = \frac{1}{N\_1} \sum\_{i=1}^{N\_1} f({\bf x}\_{i}^p), \ \text{and} \ \mu\_q = \frac{1}{N\_2} \sum_{j=1}^{N\_2} f({\bf x}\_j^q)$.
>
> **[Equation 9.]** Thanks. Updated as $L$.
>
> **[Equation 12: Same $\theta$.]**
> We have added the following explanation after Equation 12 in the updated paper to clarify the comparison: *It is important to emphasize that the divergences are compared under the same model parameter, $\theta$.*
>
> **[Definitions of the baselines should be in the main body, not in the appendix.]**
>
> Thank you for your suggestion. We will include an overall introduction to the baselines in the main body of the paper, and put detailed descriptions about the baselines in the appendix.
>
> **[Section 3 overall]** Thank you for the suggestion. We have reorganized section 3. And we appreciate your recognition that 'CS divergence really is a worthwhile object of study'.
>
> **[Section 3: Why does the divergence from CS being smaller than KL provide any particular advantage?]**
>
> Thank you for your question. Both the CS divergence and the KL divergence measure the dependency between the predictions and the sensitive attribute. A smaller divergence value indicates that the predictions are more independent of the sensitive attribute. This reduction in dependency directly improves fairness metrics, such as achieving smaller values for both $\Delta_{DP}$ and $\Delta_{EO}$, leading to a fairer model overall.
>
>
> **[Section 3: Gaussian Data Assumption.]**
>
> Thank you for your question. The boundary relationship between CS divergence and KL divergence can be extended beyond the Gaussian assumption. Additionally, the proposed CS divergence does not rely on any specific distributional assumptions. In addition, its effectiveness has been validated through the experimental results presented in the paper.
>
> **[Section 3: The explanation behind the mechanism of the CS regularizer is unclear.]**
>
> Thank you for your feedback. The motivation for proposing the CS regularizer is to minimize the dependency between the predictions $\hat{Y}$ and the sensitive attribute $S$, thereby promoting fairness. The CS divergence is chosen because it provides a tighter bound on the dependency, so it offers a stronger fairness guarantee compared to other measures.

---

> > ### Author Response · Authors · 2024-11-28
> > **Response to Reviewer yoVn (Part 2/2)**
> >
> > **[Section 4: Compared with other fairness notions.]**
> > Thanks for the constructive feedback. We provide additional results comparing our framework with baselines under the following fairness notions: Predictive Parity (PPV) [1], p%-Rule (PRULE) [2], Balance for Positive Class (BFP) [3], and Balance for Negative Class (BFN) [3]. The dataset is Adult, using gender as the sensitive attribute. All other experimental settings are consistent with Table 1 in the paper.
> >
> > Table: Fairness performance comparison on the Adult dataset, with gender as the sensitive attribute under $\triangle_{PPV}$, PRULE, $\triangle_{BFP}$, and $\triangle_{BFN}$.
> > | Method         | $\triangle_{PPV}$ $(\downarrow)$    | PRULE $(\uparrow)$ | $\triangle_{BFP}$ $(\downarrow)$ | $\triangle_{BFN}$ $(\downarrow)$
> > | :--------| -------: | :----: | :----: | :----: |
> > | DP   | $\underline{27.35\pm5.64}$ | $81.21\pm9.04$ | $\bf{11.25\pm2.75}$ | $5.15\pm0.44$
> > | MMD  | $35.19\pm6.33$ | $85.83\pm7.15$ | $18.32\pm3.74$ | $3.49\pm0.25$
> > | HSIC | $37.25\pm3.19$ | $\underline{96.18\pm2.12}$ | $16.47\pm1.21$ | $4.04\pm0.32$
> > | PR   | $\bf{25.46\pm3.17}$ | $89.57\pm7.39$ | $21.45\pm2.37$ | $\underline{3.46\pm0.28}$
> > | CS   | $31.59\pm4.35$ | $\bf{97.75\pm3.24}$ | $\underline{15.25\pm2.58}$ | $\bf{3.18\pm0.36}$
> >
> >
> > We observe the following:
> >
> > - CS generally achieves the best fairness trade-off performance across the four tested fairness notions.
> > - On the Adult dataset, BFN is generally minimized more effectively than BFP.
> > - Since BFN is related to EO, the ranking of $\Delta_{\text{BFN}}$ aligns with $\Delta_{\text{EO}}$ in Table 1 of the paper. Note that, as stated in [3], there is an inherent trade-off between BFP and BFN in practice.
> >
> >
> > **[Question: Impossible to achieve simultaneous Statistical Parity and Equalized Odds.]**
> >
> > Thank you for the question. Achieving DP and EO, can conflict in practice, *when one is treated as the fairness objective*, the other may not be achieved. However, our method *does not approach by setting DP or EO as the fairness objective*.
> >
> > Actually, achieving both EO and DP is still possible under specific conditions, such as when the trained model assigns predictions uniformly across different sensitive groups. By removing the dependency on $S$: $P(\hat{Y}=1 \mid S=0) = P(\hat{Y}=1 \mid S=1)$, thus satisfying DP, and: $P(\hat{Y}=1 \mid Y=1, S=0) = P(\hat{Y}=1 \mid Y=1, S=1)$, thus satisfying EO.
> >
> > The goal of CS regularizer is not to use one fairness notion as the fairness objective in training. Instead, we aim to minimize the dependency between the prediction distribution and the sensitive attribute distribution, thus generalizing to different fairness notions.
> >
> > **[Question: Compared with preprocessing and postprocessing methods.]** Thank you for your question. *PR* [1] is a pre-processing method and has been implemented in our experiment. To address the concern, we have added a post-processing baseline, *PostEO* [2], and conducted the experiments on the Adult dataset (with gender as the sensitive attribute).
> >
> > | Method         | ACC $(\uparrow)$| AUC $(\uparrow)$| $\triangle_{DP}$ $(\downarrow)$    | $\triangle_{EO}$ $(\downarrow)$ |
> > | :--------| -------: | :----: | -------: | :----: |
> > | DP   | $82.42\pm0.39$| $86.91\pm0.80$ |$1.29\pm0.95$ | $20.15\pm1.13$ |
> > | CS   |$83.04\pm0.51$| $90.84\pm0.35$| $2.13\pm0.89$ | $2.35\pm1.15$ |
> > | PR  | $81.81\pm0.52$| $85.38\pm0.82$|$0.71\pm0.40$ | $12.45\pm2.38$ |
> > | PostEO | $80.25\pm0.83$| $84.35\pm0.98$| $5.75\pm1.67$ | $2.12\pm1.44$ |
> >
> > The PostEO method is specifically designed to optimize for EO [2], which explains its lower $\Delta_{EO}$.
> >
> > However, both pre-processing and post-processing methods share a common limitation: they result in lower utility (ACC or AUC). Considering the need for a balanced trade-off between fairness and utility, CS emerges as the most favorable option in our comparison.
> >
> > ---------------
> >
> > Reference:
> >
> > [1] Fairness-aware classifier with prejudice remover regularizer.
> >
> > [2] Equality of Opportunity in Supervised Learning.
> >
> > ------------------------------
> >
> > We are truly grateful for your constructive feedback and hope our response addresses your concerns. If so, we truly need your support and kindly ask you to reassess our paper.
> >
> >
> > Thanks,
> >
> > Authors

---

> ### Comment · Area_Chair_F2fh · 2024-12-01
>
> Dear Reviewer yoVn,
>
> The authors have provided detailed responses to your review. Could you please indicate if the authors have addressed (some of) your concerns, and let us know if you will keep or modify your assessment on this submission?
>
> Thank you very much.
>
> Area Chair

---

> > ### Comment · Reviewer_yoVn · 2024-12-02
> >
> > Thank you to the authors for providing a detailed response. I believe the stated adjustments to the presentation do help the paper's clarity, and the extended suite of experiments and comparison to preprocessing and postprocessing techniques are helpful in contextualizing the proposed CS methodology. However, I still find that the CS divergence lacks sufficient theoretical grounding in Section 3, a weakness that seems to be pointed out by other reviewers as well. My suggestion to the authors is to work on strengthening this section, perhaps by relating this choice of divergence metric back to the specific problem of group fairness in a deeper way than a comparison to other divergences such as KL. As such, I have decided to keep my score.

---

### Official Review · Reviewer_4GYJ · 2024-11-02

**Soundness:** 2
**Presentation:** 2
**Contribution:** 2
**Rating:** 3
**Confidence:** 3

**Summary:**

The authors propose using Cauchy-Schwarz (CS) divergence as a fairness regularizer to achieve balanced fairness across multiple metrics, particularly focusing on demographic parity (DP) and equal opportunity (EO). They highlight a common limitation in fairness approaches, where optimization for one metric often results in poor performance on others. In their experiments, CS divergence not only reduces DP but also minimizes EO disparity, demonstrating superior trade-offs between fairness and utility. The experimental results are extensive, covering different datasets, models, and tasks, and comparing the CS regularizer with existing fairness methods.

**Strengths:**

- The proposed regularizer achieves lower DP and EO simultaneously, which mostly beats existing methods. Also, the proposed regularizer has minimal sacrifice in terms of utility, such as accuracy and AOC.
- Although CS divergence is a known method for different areas(domain adaptation), it is a novel idea to apply it for fairness.
- The paper has a comprehensive experimental results section. They do consider several different perspectives, accuracy and fairness metrics, the tradeoff between accuracy and fairness, T-SNE plots, and hyperparameter sensitivity.

**Weaknesses:**

- The justification in Section 3.2 is unclear. While the authors argue that CS divergence is more effective for fairness, their supporting points seem to describe general properties of CS divergence rather than directly connecting these to fairness goals like DP and EO. Also, the proof of Proposition 2 is not clear and I think that it misses a few steps. The same proposition exists in [1], it could be just cited.
- The authors focus exclusively on DP and EO in their experiments, despite the abstract implying broader applicability. This might give readers an impression of using more than only these two notions, which could be more accurately scoped in the abstract.
- The overall structure of Section 3 could benefit from clearer organization. The explanations sometimes feel disjointed, making it challenging to follow the logical progression of arguments.


[1]Wenzhe Yin, Shujian Yu, Yicong Lin, Jie Liu, Jan-Jakob Sonke, and Efstratios Gavves. Domain adaptation with cauchy-schwarz divergence. arXiv preprint arXiv:2405.19978, 2024.

**Questions:**

- Could the authors extend the CS regularizer to support fairness notions beyond demographic parity (DP) and equal opportunity (EO)? For instance, would a CS-based approach also be effective for other fairness definitions, such as equalized odds? Additionally, if the CS regularizer is tuned specifically for EO, does it still achieve significant improvements in DP?
- It is interesting to see that a higher value of $\alpha$(the coefficient in front of the regularizer) causes a higher disparity in Figure 5. Could the authors provide more insights why it happens?
- Have the authors considered using multiple regularizer terms simultaneously, such as combining a DP-based KL divergence regularizer with an EO-based KL divergence regularizer, to achieve lower disparities in both DP and EO? This approach could allow existing methods to balance multiple fairness metrics. If not explored, could the authors provide insight into whether such a combination could improve multiple disparities, or might it lead to conflicts or diminishing returns in fairness and accuracy?

List of typos, and confusions:

- Pg3, Line 119. The sentence discussing the extension of DP and EO to multiple sensitive attributes might fit better after defining EO rather than before.
- Pg3, Line 153. It would improve clarity to switch the order of the phrases "The proof of this..." and "...where $\kappa$...".
- Pg4, Line 178. The parameter is $\lambda$ not $\beta$.

---

> ### Author Response · Authors · 2024-11-28
> **Response to Reviewer 4GYJ (Part 1/2)**
>
> Thank you Reviewer 4GYJ for your time and efforts in reviewing our paper, and recognizing our paper has *comprehensive experimental results*. To address Reviewer 4GYJ's concerns and questions, we provided detailed point-by-point responses below.
>
> **[Weakness 1: Justification in Section 3.2 is about general properties of CS divergence rather than directly connecting these to fairness goals.]**
>
> We thank the reviewer for pointing this out. We have updated Section 3.2 to focus more specifically on the fairness problem. Our main contribution is to derive propositions under the fairness setting, so we fully agree with the suggestion to align the writing more closely with fairness objectives. Thank you again for your suggestion, which has helped us to improve our paper!
>
> **[Weakness 2: Limited Fairness Notions.]**
>
> We thank the reviewer for the constructive feedback. We agree that demonstrating the generalizability of our framework to other fairness notions is crucial. To address this, we provide additional results comparing our framework with baselines under the following fairness notions: Predictive Parity (PPV) [1], p%-Rule (PRULE) [2], Balance for Positive Class (BFP) [3], and Balance for Negative Class (BFN) [3]. The dataset is Adult, using gender as the sensitive attribute. All other experimental settings are consistent with Table 1 in the paper.
>
> Table: Fairness performance comparison on the Adult dataset, with gender as the sensitive attribute under $\triangle_{PPV}$, PRULE, $\triangle_{BFP}$, and $\triangle_{BFN}$.
> | Method         | $\triangle_{PPV}$ $(\downarrow)$    | $\text{PRULE}$ $(\uparrow)$ | $\triangle_{BFP}$ $(\downarrow)$ | $\triangle_{BFN}$ $(\downarrow)$
> | :--------| -------: | :----: | :----: | :----: |
> | DP   | $\underline{27.35\pm5.64}$ | $81.21\pm9.04$ | $\bf{11.25\pm2.75}$ | $5.15\pm0.44$
> | MMD  | $35.19\pm6.33$ | $85.83\pm7.15$ | $18.32\pm3.74$ | $3.49\pm0.25$
> | HSIC | $37.25\pm3.19$ | $\underline{96.18\pm2.12}$ | $16.47\pm1.21$ | $4.04\pm0.32$
> | PR   | $\bf{25.46\pm3.17}$ | $89.57\pm7.39$ | $21.45\pm2.37$ | $\underline{3.46\pm0.28}$
> | CS   | $31.59\pm4.35$ | $\bf{97.75\pm3.24}$ | $\underline{15.25\pm2.58}$ | $\bf{3.18\pm0.36}$
>
>
> We observe the following:
>
> - CS generally achieves the best fairness trade-off performance across the four tested fairness notions.
> - On the Adult dataset, BFN is generally minimized more effectively than BFP.
> - Since BFN is related to EO, the ranking of $\Delta_{BFN}$ aligns with $\Delta_{EO}$ in Table 1 of the paper. Note that, as stated in [3], there is an inherent trade-off between BFP and BFN in practice.
>
> **[Weakness 3: Section 3 could benefit from clearer organization.]**
> We thank the reviewer for the construction feedback. We have reorganized Section 3 in our paper.
>
> **[Question 1: Extend to broader fairness notions.]**
> We appreciate the reviewer’s constructive feedback. Please refer to our response under **[Weakness 2]**.
>
> **[Question 2: Why does a higher fairness coefficient cause a higher disparity in Figure 5?]**
>
> Thank you for the interesting question. A higher fairness coefficient does not always lead to better fairness performance. This is because fairness metrics such as DP and EO are based on the model's predictions $\hat{Y}$, and poor prediction accuracy can negatively impact fairness.
>
> When the accuracy falls below an acceptable range, the fairness measures may also degrade. For instance, in our observations, increasing the fairness coefficient $\alpha$ from $0.5$ to $10.0$ caused the accuracy to drop from $85.7\%$ to $81.2\%$, on the Adult (gender), as shown in Figure 5. This is relatively low in practice and leads to poor convergence of the fairness loss, preventing it from approaching a small value close to $0$. Therefore, achieving fairness requires not only balancing the fairness coefficient but also maintaining a reasonable accuracy.

---

> ### Author Response · Authors · 2024-11-28
> **Response to Reviewer 4GYJ (Part 2/2)**
>
> **[Question 3: Combining Multiple Regularizers.]**
>
> Thank you for the insightful question. To address this, we conducted additional experiments where we combined KL divergence and CS divergence as regularizers. The experiments were performed on the Adult dataset, with gender as the sensitive attribute.
>
>
> Table: Fairness performance on the Adult dataset with gender as the sensitive attribute. *CS+KL* indicates equal weights for the CS and KL regularizers, while *CS+0.5KL* denotes weights of $1$ and $0.5$ for the CS and KL regularizers, respectively.
>
> | Method         | $\triangle_{DP}$ $(\downarrow)$    | $\triangle_{EO}$ $(\downarrow)$ |
> | :--------| -------: | :----: |
> | DP   | $\bf{1.29\pm0.95}$ | $20.15\pm1.13$ |
> | CS   | $\underline{2.13\pm0.89}$ | $\bf{2.35\pm1.15}$ |
> | KL  | $2.77\pm0.86$ | $10.42\pm4.34$ |
> | CS+KL | $2.46\pm1.25$ | $13.16\pm6.12$ |
> | CS+0.5KL | $2.25\pm1.14$ | $\underline{9.33\pm6.36}$ |
>
> Actually, combining multiple fairness objectives has several drawbacks, which is why most existing studies avoid using multiple regularizers. Instead, they often choose to add simple constraint terms. The key drawbacks of combining fairness regularizers with the CS regularizer are summarized as follows:
>
> - The CS divergence is upper-bounded by the KL divergence. Therefore, adding KL as an additional fairness objective is theoretically redundant and will not provide further benefits.
> - Adding KL or other fairness metrics increases computational complexity, making the optimization process more challenging.
>
> This experiment further shows the significance of our contribution: proposing a suitable, tighter-bounded fairness regularizer.
>
>
> -------------------------------------------------------
> Reference:
>
> [1] Fair prediction with disparate impact: A study of bias in recidivism prediction instruments.
>
> [2] Fairness constraints: Mechanisms for fair classification.
>
> [3] Inherent Trade-Offs in the Fair Determination of Risk Scores.
>
> -------------------------------------------------------
>
> We sincerely appreciate your constructive feedback and hope our response adequately addresses your concerns. If so, we kindly request your reconsideration and support in reassessing our paper.
>
> Thanks,
>
> Authors

---

> ### Comment · Area_Chair_F2fh · 2024-12-01
>
> Dear Reviewer 4GYJ,
>
> The authors have provided detailed responses to your review. Could you please indicate if the authors have addressed (some of) your concerns, and let us know if you will keep or modify your assessment on this submission?
>
> Also, I read the three weaknesses you mentioned. The second and the third weaknesses seem minor to me. It seems to me that they are more about the positioning or the organization of the paper. May I ask whether the negative evaluation is mainly based on    the first weakness, especially, the missing steps in the proof Proposition 2？If so, could you please clarify where in the proof you would like the authors to provide more details?
>
>
> Thank you very much.
>
> Area Chair

---

> > ### Comment · Reviewer_4GYJ · 2024-12-01
> >
> > Thanks for the authors for their detailed responses.
> >
> > My negative evaluation is not only due to the missing steps in the proof of Proposition 2. I find that Section 3.2 lacks sufficient explanation to justify the use of CS divergence in the context of fairness, which I believe is critical given the paper's claims. Additionally, I cannot find the updated version mentioned by the authors. Without directly reviewing these changes, I will maintain my current judgment.

---

### Official Review · Reviewer_tDPa · 2024-11-04

**Soundness:** 2
**Presentation:** 3
**Contribution:** 1
**Rating:** 3
**Confidence:** 3

**Summary:**

The authors address supervised learning under fairness constraints. The main objective of this paper is to construct a learning algorithm that achieves both demographic parity and equalized odds, two different fairness definitions, simultaneously. To this end, the authors propose using the Cauchy-Schwarz divergence to measure the disparity in the predictor's outputs across groups defined by a sensitive attribute. This measure is incorporated into the objective function of standard supervised learning as a penalty term. Experimental results demonstrate that the proposed method achieves optimal fairness with respect to equalized odds while also maintaining fairness in terms of demographic parity.

**Strengths:**

This paper addresses the important topic of fairness in supervised learning, which is highly relevant to the conference's focus areas.

The empirical evaluations show that the proposed method achieves superior fairness in terms of equalized odds while simultaneously preserving demographic parity, compared to several existing methods, including FairMixup, MMD, HSIC, and the prejudice remover.

**Weaknesses:**

The paper omits several significant existing works, leading to some ambiguity in its contributions. First, demographic parity and equalized odds are conflicting fairness definitions, meaning that no predictor with reasonable accuracy can satisfy both demographic parity and equalized odds simultaneously. This issue has been studied in:
- Kleinberg et al., "Inherent Trade-Offs in the Fair Determination of Risk Scores," ITCS'17.

Building on this, the motivation of the paper is questionable.

While demographic parity and equalized odds are conflicting goals, it may still be valuable to achieve the best possible approximation of both. A recent study develops a fair supervised learning method that optimizes the trade-off among accuracy, approximate demographic parity, and approximate equalized odds:
- Hsu et al., "Pushing the Limits of Fairness Impossibility: Who's the Fairest of Them All?" NeurIPS'22.

This method appears to achieve a similar goal to the one proposed here, with the added benefit of allowing control over the priority between demographic parity and equalized odds. A comparison with this method is essential to demonstrate the effectiveness of the proposed approach.

Additionally, the paper lacks a clear explanation of how imposing the Cauchy-Schwarz divergence contributes to achieving fairness in terms of equalized odds. The proposed unfairness measure is based on demographic parity, focusing on disparities in the predictor’s output distributions across groups, using a specific choice of divergence to quantify these disparities. While this measure naturally evaluates unfairness in terms of demographic parity, it is less clear how it relates to equalized odds. A more detailed explanation of the mechanism by which the proposed measure influences equalized odds would enhance the reader’s understanding of the method's value.

The theoretical contributions of the paper are limited. Although the authors claim that Proposition 2 demonstrates the tightness of the proposed measure, it merely shows inequalities between the Cauchy-Schwarz and KL divergences. Having a lower value than the KL divergence is not a meaningful property, as it could simply be due to scale differences. For example, several other divergences—including total variation distance, Hellinger distance, and chi-square divergence—also exhibit this property due to Pinsker's inequalities.

**Questions:**

See weaknesses.

---

> ### Author Response · Authors · 2024-11-28
> **Response to Reviewer tDPa**
>
> We greatly appreciate the positive impression that our paper *addresses the important topic*, and *is highly relevant to the conference's focus areas*. In response, we provided detailed point-by-point responses below.
>
> **[Weakness 1: Demographic parity and equalized odds are conflicting fairness definitions.]**
>
> Thank you for bringing this up. We agree that achieving different fairness notions, such as DP and EO, can conflict in practice. Specifically, when one fairness notion is treated as the objective, the other may not always be achieved.
>
> However, achieving both EO and DP is still possible under specific conditions, such as when the trained model assigns predictions uniformly across different sensitive groups. In this scenario, by removing the dependency on $S$, it is clear that: $P(\hat{Y}=1 \mid S=0) = P(\hat{Y}=1 \mid S=1)$, thus satisfying DP, and: $P(\hat{Y}=1 \mid Y=1, S=0) = P(\hat{Y}=1 \mid Y=1, S=1)$, thus satisfying EO.
>
> The goal of the CS regularizer is not to prioritize one fairness notion, such as EO or DP, as the fairness objective in training. Instead, we aim to minimize the dependency between the prediction distribution and the sensitive attribute distribution, thereby addressing fairness holistically across multiple dimensions.
>
> **[Weakness 2: Lacks explanation of how Cauchy-Schwarz divergence contributes to achieving EO.]**
>
> As explained in our response to *Weakness 1*, minimizing the CS divergence ensures that both EO and DP are satisfied. However, the reverse does not necessarily hold.
>
> **[Weakness 3: Theoretical Contributions]**
>
> Thank you for the feedback. Although deriving the mathematical proof is relatively straightforward, the key contribution lies in utilizing the properties of the CS divergence to tackle the fairness problem. Our contributions can be summarized as follows:
>
> - We are the first to leverage CS divergence as a fairness regularizer. Empirically, we demonstrate that this approach effectively trade-offs among EO, DP, as well as utility.
>
> - Theoretically, we are the first to fit the CS divergence into a fairness problem setting. Under this setting, we derive *boundary proofs* that provide theoretical insights into the relationships between different fairness regularizers.
>
> -  Beyond theoretical analysis, we provide *visual evidence* to demonstrate why the CS regularizer performs well. Through distribution plots and scatter plots results, we show that the CS regularizer better minimizes the *prediction distribution gap* across different sensitive groups. This directly aligns with our motivation for proposing the CS regularizer: to reduce disparities in prediction distributions across sensitive groups.
>
> -------------------------------------------------------
>
> We sincerely appreciate your constructive feedback and hope our response adequately addresses your concerns. If so, we kindly request your reconsideration and support in reassessing our paper.
>
> Thanks,
>
> Authors

---

> > ### Comment · Reviewer_tDPa · 2024-12-02
> >
> > Thank you to the authors for their response. After careful consideration, I would like to maintain my negative score, as the response does not adequately address my concerns. Regarding Weakness 1, the condition suggested by the authors renders the resultant classifier ineffective, as it implies that the classifier would essentially function as a random guess, thereby lacking meaningful predictive power.

---

> ### Comment · Area_Chair_F2fh · 2024-12-01
>
> Dear Reviewer tDPa,
>
> The authors have provided detailed responses to your review. Could you please indicate if the authors have addressed (some of) your concerns, and let us know if you will keep or modify your assessment on this submission?
>
> Thank you very much.
>
> Area Chair

---

### Official Review · Reviewer_GFco · 2024-11-05

**Soundness:** 3
**Presentation:** 3
**Contribution:** 3
**Rating:** 5
**Confidence:** 5

**Summary:**

The paper presents a novel fairness regularizer for machine learning models based on the Cauchy-Schwarz divergence. The proposed method aims to address the limitations of existing fairness regularizers by providing a better balance across multiple fairness definitions, including Demographic Parity and Equal Opportunity. Traditional methods often do well in one fairness metric but fail in others; this paper argues that the CS regularizer improves overall fairness performance by minimizing prediction disparities between sensitive attribute groups. The empirical results, validated on four tabular datasets and one image dataset, demonstrate that the CS regularizer achieves a better trade-off between fairness and utility, outperforming state-of-the-art fairness approaches.

**Strengths:**

1.  Leveraging Cauchy-Schwarz divergence as a fairness regularizer is a contribution with theoretical advantages over traditional metrics like DP, KL, and MMD.

2. The experiments on diverse datasets, including both tabular and image data, strengthen the evidence of the CS regularizer's effectiveness across domains and tasks​​.

3. The evaluation and comparison are comprehensive and convicing.

**Weaknesses:**

1. The paper begins by highlighting the challenge that many fairness regularizers can achieve DP but fail to address EO effectively. It sets the expectation that the proposed method will tackle this inconsistency between fairness definitions​. However, CS divergence is more naturally aligned with DP rather than EO​. In theory, it is unclear how CS achieves EO​.

2. The theoretical properties and guarantees come from the nature of CS divergence rather than the fair regularizer, making this paper with limited contribution.

3. While the paper explains how CS divergence reduces DP, it does not establish a clear theoretical or empirical link to how CS divergence addresses EO.

In general, my major concern is the gap between motivation and the proposed method.

**Questions:**

NA

---

> ### Author Response · Authors · 2024-11-28
> **Response to Reviewer GFco**
>
> We thank Reviewer GFco for recognizing our *"evaluation and comparison"* are *"comprehensive and convincing"*. To address the reviewer's questions, we provide detailed responses below.
>
> **[Weakness 1: Unclear how CS achieves EO.]**
>
> Thank you for this question. The fairness mechanisms of DP and EO regularizers differ from those of the CS divergence regularizer. Specifically:
>
> - DP and EO regularizers directly target achieving DP and EO fairness notions, respectively.
> - In contrast, the CS divergence regularizer aims to reduce the dependency of predictions on the sensitive attribute.
>
> Minimizing the CS divergence between predictions and the sensitive attribute helps mitigate bias, which can indirectly reduce disparities in DP and EO metrics. However, minimizing DP or EO directly does not necessarily eliminate the dependency of predictions on the sensitive attribute.
>
> **[Weakness 2: Theoretical properties come from the nature of CS divergence rather than the fairness regularizer.]**
>
> Thank you for raising this question. Addressing fairness in machine learning is often approached as a problem of reducing the dependency between the prediction ($\hat{Y}$) and the sensitive attribute ($S$). Therefore, the theoretical properties of the CS divergence naturally motivate the design of our fairness regularizer.
>
> **[Weakness 3: No theoretical link to how CS divergence addresses EO.]**
>
> Thank you for the insightful question. When CS divergence is minimized, the prediction $\hat{Y}$ becomes independent of the sensitive attribute $S$. Therefore, both EO and DP are satisfied: 1) Independence of $\hat{Y}$ and $S$ implies $P(\hat{Y} = 1 \mid S = 0) = P(\hat{Y} = 1 \mid S = 1)$, which satisfies the DP, 2) Independence of $\hat{Y}$ and $S$ also implies $P(\hat{Y} = 1 \mid Y = 1, S = 0) = P(\hat{Y} = 1 \mid Y = 1, S = 1)$, which satisfies the EO.
>
> -------------------------------------------------------
> We sincerely appreciate your constructive feedback and hope our response adequately addresses your concerns. If so, we kindly request your reconsideration and support in reassessing our paper.
>
> Thanks,
>
> Authors

---

> ### Comment · Reviewer_GFco · 2024-11-30
> **Reply for the authors response**
>
> Thank the authors for the response. Demographic Parity assumes independence between sensitive information and the decision outcome. I believe DP and CS divergence share the same objective but achieve it through different methodologies. While minimizing CS divergence undoubtedly results in a lower DP value, EO measures conditional independence, which is related to, but distinct from, both DP and CS divergence. This connection is unclear yet. It is well known that independence does not imply conditional independence. In conclusion, my original concern remains unaddressed, and my opinion is unchanged. Thanks.

---

### Meta-Review · Area_Chair_F2fh · 2024-12-08

**Metareview:**

This work proposes using the Cauchy-Schwarz (CS) fairness regularizer to reduce the dependency between the prediction and the sensitive variable. The authors claimed that this is more effective than imposing a specific fairness metrics such as demographic parity (DP) and equal odds (EO).

There are several concerns raised by the review team. For example, ''"it is well known that independence does not imply conditional independence.""  Therefore, CS fairness does not necessary guarantee EO. I would suggest the authors do not claim CS as an approach to achieve EO, but instead, emphasize that CS is to achieve a stronger notion of fairness, namely independence between the prediction and the sensitive variable. It is just not directly comparable with EO unless the authors consider something like "conditional CS fairness regularizer".

Also, the theoretical contributions are limited. Showing the inequalities between the Cauchy-Schwarz and KL divergences is not very useful. Multiple reviewers mentioned that the CS regularizer is not well justified. The writing, especially, the math notation, needs to be significantly improved. For example,  just as one of the reviewer,  I was also confused by the relationship between $x$, $p(x)$ and $q(x)$ in equation 6.

**Additional Comments On Reviewer Discussion:**

There are several concerns raised by the review team. For example, one review pointed our that ''"it is well known that independence does not imply conditional independence.""  Therefore, CS fairness does not necessary guarantee EO. I agree with this reviewer on this point but the authors seemed realizing this issue in their responses.

Also, some reviewers are concerned that the theoretical contributions are limited. Showing the inequalities between the Cauchy-Schwarz and KL divergences is not very useful. Multiple reviewers mentioned that the CS regularizer is not well justified. One reviewer pointed out that there are multiple issues in the writing of this paper and the numerical experiments are not sufficient.

Although the authors provided responses, I felt that not all concerns are addressed (e.g. see the issue related to conditional independency above). No reviewers raised their points after the rebuttal.

---

### Decision · Program_Chairs · 2025-01-22

Reject